# Pre-hypertrophic chondrogenic enhancer landscape of limb and axial skeleton development

Fabrice Darbellay [1,2,3], Anna Ramisch [4], Lucille Lopez-Delisle [5], Michael Kosicki[3], Antonella Rauseo[1,2], Zahra Jouini [1,2], Axel Visel [3,6,7] & Guillaume Andrey [1,2] ✉

Chondrocyte differentiation controls skeleton development and stature. Here we provide a comprehensive map of chondrocyte-specific enhancers and show that they provide a mechanistic framework through which non-coding genetic variants can influence skeletal development and human stature. Working with fetal chondrocytes isolated from mice bearing a *Col2a1* fluorescent regulatory sensor, we identify 780 genes and 2'704 putative enhancers specifically active in chondrocytes using a combination of RNA-seq, ATAC-seq and H3K27ac ChIP-seq. Most of these enhancers (74%) show *pan*-chondrogenic activity, with smaller populations being restricted to limb (18%) or trunk (8%) chondrocytes only. Notably, genetic variations overlapping these enhancers better explain height differences than those overlapping non-chondrogenic enhancers. Finally, targeted deletions of identified enhancers at the *Fgfr3*, *Col2a1*, *Hhip* and, *Nkx3-2* loci confirm their role in regulating cognate genes. This enhancer map provides a framework for understanding how genes and non-coding variations influence bone development and diseases.

Chondrocytes and their extracellular matrix constitute the building blocks of fetal cartilage modules, which are gradually modified into bones through endochondral ossification[1]. Mutations in transcription factors (SOX9), extracellular matrix (COL10A1, COL2A1) or paracrine signaling (FGFR3) proteins involved in chondrocyte differentiation have been shown to affect height but also induce a wide variety of skeletal disorders[2,3]. Specifically, the role of chondrocytes in endochondral ossification is complex and involves a precise developmental trajectory. First, mesenchymal cells differentiate into chondrocytes, which then undergo further steps of stratified differentiation[1,4]. In the central strata of condensations, chondrocytes become hypertrophic and enable mineral deposition, driving the calcification of the developing bones. In contrast, near the ends of growing bones,

chondrocytes form a growth plate that sustains the elongation of bones and ultimately determines the stature of individuals[3]. Although initial sets of chondrogenic genes have been identified and the function of many of them studied in detail[1], the regulatory architecture that controls their activity during development remain mostly unknown.

Distant-acting enhancers are a key component of the gene regulatory architecture that orchestrates the spatiotemporal and cell-specific transcriptional activities of their target genes[5]. To do so, they integrate regulatory input from signaling pathways and cellular states over time through the binding of transcription factors. Developmental genes are often controlled by several enhancers that collectively enable their complex expression patterns and act in a functionally redundant manner[6,7]. On the one hand, multiple enhancers are often

[1]Department of Genetic Medicine and Development, Faculty of Medicine, University of Geneva, 1211 Geneva, Switzerland. [2]Institute of Genetics and Genomics in Geneva (iGE3), University of Geneva, 1211 Geneva, Switzerland. [3]Environmental Genomics and Systems Biology Division, Lawrence Berkeley Laboratory, Berkeley, CA 94720, USA. [4]Department of Basic Neurosciences, Faculty of Medicine, University of Geneva, 1211 Geneva, Switzerland. [5]School of Life Sciences, Ecole Polytechnique Fédérale de Lausanne (EPFL), 1015 Lausanne, Switzerland. [6]U.S. Department of Energy Joint Genome Institute, Lawrence Berkeley Laboratory, Berkeley, CA 94720, USA. [7]School of Natural Sciences, University of California, Merced, CA 95343, USA. ✉e-mail: guillaume.andrey@unige.ch

found to be essential to express a given gene in a specific organ or tissue[5]. On the other hand, developmental genes are often pleiotropic, active in several tissues, and thus rely on distinct enhancer repertoires for each tissue they are expressed in[7]. To achieve their function, developmental genes thus rely on different enhancer types, ranging from early patterning enhancers to late cell-type specific enhancers[8], collectively ensuring that a gene is expressed in the right cell type, over defined durations at diverse embryonic positions.

A comprehensive chondrogenic enhancer inventory would define the regulatory landscape of genes involved in chondrogenesis and enable the interpretation of non-coding sequence variants associated with variation in skeletal morphology and rare congenital bone pathologies. Using binding sites of the chondrogenic transcription factor SOX9 in postnatal dissected mouse as a proxy, initial sets of SOX9-dependent rib chondrocyte enhancers were identified[9]. Likewise, chromatin profiling has been used to identify candidate enhancers active during the in vitro differentiation of human mesenchymal stem cells into chondrocytes[10]. Finally, the open chromatin signatures of chondrocytes in vivo were mapped to characterize their epigenetic landscape, yet, with limited specificity when it comes to identifying active chondrogenic enhancer regions as such a signature alone both marks poised and active regions[11,12]. Therefore, despite these efforts, the genome-wide landscape of enhancers active in chondrocytes in vivo during prenatal development of the axial skeleton and long bones of the extremities, which are largely responsible for stature and commonly affected by rare skeletal disorders, remains undefined. In the present study, we identify enhancers active in chondrocytes isolated from embryonic skeletal elements taking part in endochondral ossification in vivo.

Developmental enhancers can be identified by analysis of epigenomic signatures from microdissected bulk tissues, revealing characteristic chromatin states associated to enhancers. Since tissues contain multiple cell types, the ability to define cellular specificity of enhancers through this approach is limited. This challenge can be overcome through the analysis of activities in sorted cell populations isolated from complex developing tissues[13]. To map chondrogenic enhancers in vivo we devise a fluorescent regulatory sensor approach to isolate chondrocytes from fetuses and characterize their transcriptomic and chromatin landscapes. By using a combination of accessible chromatin profiling and enhancer-associated mark H3K27ac we map and characterize a genome-wide set of chondrogenic enhancers active during limb and trunk development[14].

## Results

### A regulatory sensor enables isolation of fetal chondrocytes

To enable the isolation of chondrocytes from embryonic tissues in sufficient quantities to perform enhancer mapping by ChIP-seq, we engineered a fluorescent reporter mouse line. Chondrocytes are a heterogenous cell population characterized by different levels of cell maturity and extracellular matrix secretion[1]. To identify a broad range of chondrocytes, we selected the *Col2a1* locus to drive expression of an EGFP marker gene. *Col2a1* expression is present in early to pre-hypertrophic chondrocytes, a progenitor type of cells involved in the formation of bones[1,4]. We established a homozygous *Col2a1* reporter Embryonic Stem Cell (ESC) line carrying a regulatory sensor cassette with a minimal ß-globin promoter and an EGFP coding sequence inserted 1.2 kb upstream of the *Col2a1* transcription start site (*Col2a1$^{EGFP}$*, Fig. 1A). We then derive fetuses from *Col2a1$^{EGFP}$* ESCs using

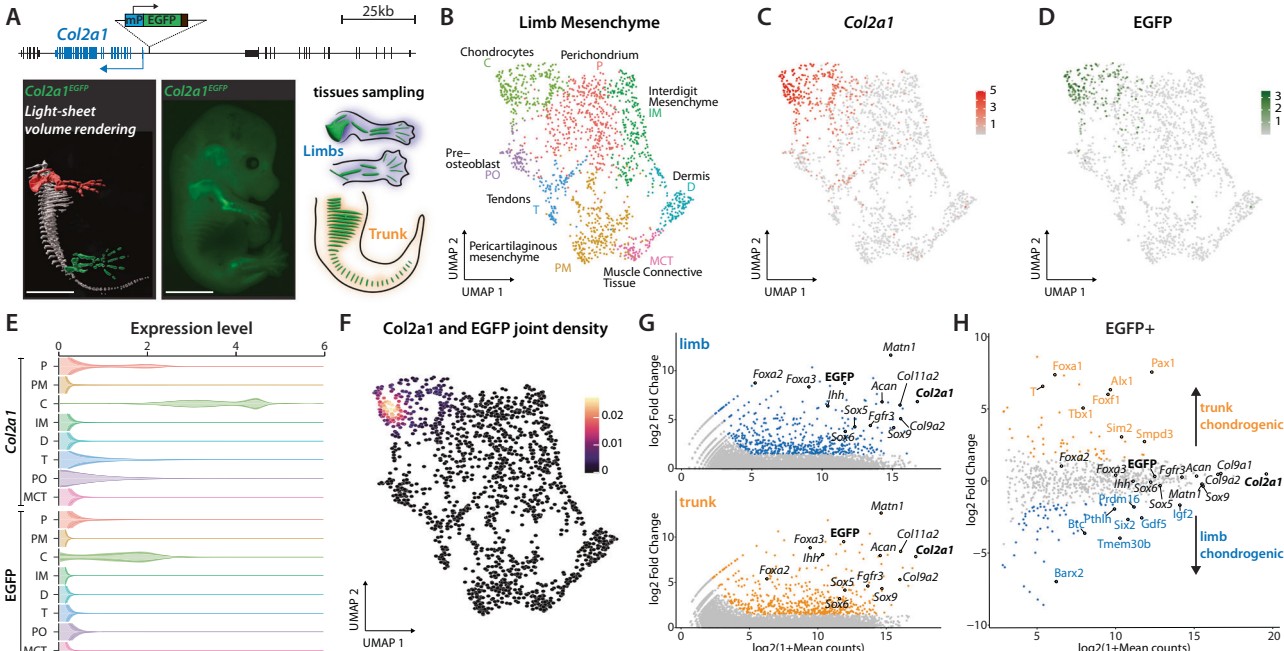

**Fig. 1 | Study overview and validation of the approach. A** Top: a fluorescent regulatory sensor cassette was integrated 1.2 kb upstream of the *Col2a1* promoter. Bottom left: light-sheet microscopy reconstruction of right half of a E14.5 *Col2a1$^{EGFP}$* fetus; center: stereomicroscope image of E14.5 *Col2a1$^{EGFP}$* fetus; right: sketch of microdissected limb and trunk tissues. In particular, stylo-, zeugo-, and autopods were harvested from fore and hindlimbs. Trunks were isolated from neck to tail and internal organs were removed prior to processing. Scale bars: 3 mm. **B** UMAP visualization of clustering of E14.5 limb mesenchymal cells reveals 8 clusters. **C, D** Expression of *Col2a1* (**C**) and EGFP (**D**) across the UMAP. **E** Expression of *Col2a1*

and *EGFP* per limb mesenchymal cluster. Note the strong expression of both genes in chondrocytes. **F** *Col2a1* and EGFP joint density displayed on top of the same E14.5 limb mesenchymal UMAP shown in (**B**). Co-expression of the two genes is specific to the chondrocyte clusters. **G** Significantly enriched genes in EGFP+ population in limbs (above, blue) and trunk (below, orange) versus EGFP- cells in these tissues. **H** Chondrogenic (EGFP + ) marker genes displaying a significant expression preference in limb (blue) or trunk (orange) EGFP+ cells. Statistical test used in G and H: DESeq2 Wald test, differential expression was scored when abs(log2FC)>1.5 and FDR-corrected by Benjamini-Hochberg method two-tailed *p*-value < 0.05.

tetraploid aggregation[15]. At stage E14.5, *Col2a1*[EGFP] fetuses displayed pronounced GFP signal in developing bone primordia as visualized by regular microscopy and light sheet imaging (Fig. 1A, Supplementary Movie 1). We observed fluorescent signals along the main body axis, in limbs, and in the face, recapitulating the canonical *Col2a1* expression pattern[1].

To verify the specificity of the regulatory sensor approach in marking *Col2a1*-expressing cells, we performed single-cell RNA-seq analysis of E14.5 *Col2a1*[EGFP] limbs. *Col2a1* and EGFP expression were restricted to a large cluster of mesenchymal cells, whereas several satellite clusters including epithelium, muscles, neurons, blood, immune cells, and endothelium showed no or minimal *Col2a1* and EGFP expression (Supplementary Fig. 1, Supplementary Data 1). We further subclustered the mesenchymal cell population and defined eight cell clusters including muscle connective tissue (MCT: *Dpt* + [16],), dermis (D: *Col7a1* + [17],), pericartilaginous mesenchyme (PM: *Ebf2* + [18],), tendons (T: *Scx* + [19],), pre-osteoblasts (PO: *Runx2* + [20],), perichondrium (P: *Foxp1* + [21],), interdigit mesenchyme (IM: *Msx1* + [22],), and chondrocytes (C: *Col2a1* + [1],) (Fig. 1B, Supplementary Fig. 2A, Supplementary Data 2). EGFP transcripts were strongly enriched in the chondrocyte cluster, where it was found co-expressed with *Col2a1* (correlation coefficient=0.74, p < 0.003 where p is the probability for the correlation coefficient to be negative, Fig. 1B–F and Supplementary Fig. 2B)[23]. We also observed weak expression of both *Col2a1* and EGFP in the perichondrium cluster, a signal that likely originates from the perichondrium chondrogenic cell layer (correlation coefficient=0.68, $p < 0.0054$)[23,24]. Taken together, these results confirm that the *Col2a1* regulatory sensor approach specifically marks pre-hypertrophic chondrocytes.

Following validation of the cell type specificity of the reporter signal, we isolated limb and trunk tissues from E14.5 fetuses since endochondral ossification in these regions is a main driver of bone formation (Fig. 1A). We used fluorescence-activated cell sorting (FACS) to isolate EGFP-positive (EGFP + ) and EGFP-negative (EGFP-) cells, which in both tissues represented ~10% and ~90% of all cells, respectively (Supplementary Fig. 3). To assess the expression of relevant marker genes, we performed RNA-seq on all four cell populations. EGFP+ cells from both tissues showed high expression of key chondrogenic marker genes including *Col2a1, Sox9, Foxa2, Foxa3, Matn* and *Fgfr3* (Fig. 1G, Supplementary Data 3). We did not observe an enrichment of perichondrium markers (*Foxp1, Foxp2, Sox11*) in our dataset, suggesting that we isolated a homogenous population of chondrocytes. In contrast, EGFP- cells showed high expression of markers of multiple non-chondrogenic cell populations such as muscle (*Myog, Pax3, Myod1, Myf6*), connective tissue (*Col3a1, Dcn, Kera*) and epithelium (*Krt14, Krt15*; Supplementary Fig. 4A). These results further reinforce that our approach enriches chondrocytes from both limb and trunk samples. Across 780 genes that showed increased expression in EGFP+ cells from limb or trunk, the majority (n = 655; 84%), were shared between both tissues. We also observed smaller numbers of genes whose overexpression was exclusive to limb (*n* = 67; 9%) or trunk chondrocytes (n = 58; 7%; Fig. 1H, Supplementary Data 3). This included genes with known functions in limb and axial skeleton development. For example, *Gdf5* showed high expression in a subset of limb chondrocytes and is essential for limb joint formation[25,26], whereas *Pax1* showed high expression in trunk chondrocytes and is known to be required for formation of the axial skeleton[27]. A comparable analysis performed on the EGFP- marker genes (n = 3'271) identified 523 trunk-specific and 75 limb-specific genes (Supplementary Fig. 4B, Supplementary Data 4). The larger number of trunk-specific non-chondrogenic genes likely reflects the higher tissue heterogeneity obtained from the trunk. Together, these data show that our reporter-driven enrichment approach enables the isolation of bona fide pre-hypertrophic chondrocytes from limb and trunk tissues and identifies more than 780 genes with expression specific to chondrocytes,

including more than 120 specific only to limb or to trunk chondrocytes.

## Limb- and trunk-enriched enhancers complement a common *pan*-chondrogenic enhancer landscape

To define putative enhancer regions active in limb and trunk chondrocytes, we mapped accessible chromatin regions using ATAC-seq and performed ChIP-seq for an active enhancer-associated chromatin modification, H3K27ac, in EGFP+ and EGFP- cells isolated from limb and trunk tissues[14,28]. In an initial analysis of ATAC-seq data in isolation we observed 112'095 regions showing chromatin accessibility in at least one of the four samples (Fig. 2A). Most of these regions were similar between EGFP+ and EGFP- cells in the respective tissue, suggesting that chondrocytes share a substantial proportion of their accessible chromatin landscape with other cell types present in the same tissue. Since ATAC-seq alone does not provide a strong indication of the activity status of a given regulatory region, we used H3K27ac ChIP-seq signal present at the 112'095 accessible chromatin regions to identify active regulatory regions (Fig. 2A). We observed H3K27ac signal in at least one of the four cell populations at 30'953 (39%) of these sites, suggesting they are active regulatory regions, enhancers or promoters, in at least one limb or trunk cell type.

To define a set of high-confidence enhancers in EGFP+ and EGFP- cells, we looked for regions that showed strong differential H3K27ac signal ( ≥ 4-fold) between sorted cell populations, in at least one of the two tissues, and excluded promoter regions. With this approach we defined 3'583 regions including 2'704 (75%) with stronger H3K27ac signal in EGFP+ cells (chondrogenic enhancers) and 879 (25%) with stronger H3K27ac signal in EGFP- cells (non-chondrogenic enhancers; Fig. 2B, Supplementary Fig. 5, Supplementary Data 5 and 6). The lower percentage of non-chondrogenic enhancers is likely due to a signal dilution effect resulting from the presence of multiple different cell types in the EGFP- fraction. Among the 2'704 chondrogenic enhancers, we observed 2'003 (74%) *pan*-chondrogenic enhancers with similar H3K27ac enrichment in limb and trunk chondrocytes. In contrast, a quarter of chondrogenic enhancers showed pronounced ( ≥ 2-fold) signal differences between limb and trunk chondrocytes, including 483 (18%) limb chondrocyte enhancers and 218 (8%) trunk chondrocyte enhancers (Fig. 2C, Supplementary Data 5). These results indicate that limb and trunk chondrocytes share a large common set of *pan*-chondrogenic enhancers, which is complemented by hundreds of chondrogenic enhancers specific to limb and trunk.

To associate these enhancers to potential target genes we assessed their colocalization within Topologically Associating Domains (TADs). Using a genome-wide set of TADs derived from Hi-C data generated from E14.5 mouse forelimb cartilage[29], 2'678 of 2'704 chondrogenic enhancers (99%) could be unambiguously assigned to 1'018 TADs containing at least one chondrogenic enhancer each. Next, we examined these TADs for the presence of chondrocyte-enriched protein-coding genes (as defined by RNA-seq). We identified 357 TADs that contain at least one chondrogenic enhancer and at least one protein-coding chondrocyte-specific gene. Collectively these regions, referred to as *chondroTADs* below, contain 478 chondrogenic genes and 1'363 chondrogenic enhancers (Supplementary Data 7–9). We also identified 661 TADs that contain a total of 1'315 chondrogenic enhancers in the absence of chondrocyte-specific genes, referred to as *chondroEnhTADs* (Supplementary Data 10 and 11). On average, *chondroTADs* contain almost two times as many chondrogenic enhancers per TAD as *chondroEnhTADs* (3.8 vs 2 per TAD; p-value = $4.68\mathrm{e}^{-23}$, Wilcoxon rank sum test, Fig. 2D). As *chondroTADs* contain both chondrogenic genes and enhancers, their functional involvement in chondrocyte differentiation and function is highly plausible. This notion is supported by Gene Ontology (GO) term analysis, indicating that protein-coding genes in *chondroTADs* (n = 4'694, Supplementary Data 12) are enriched in functions including bone and cartilage

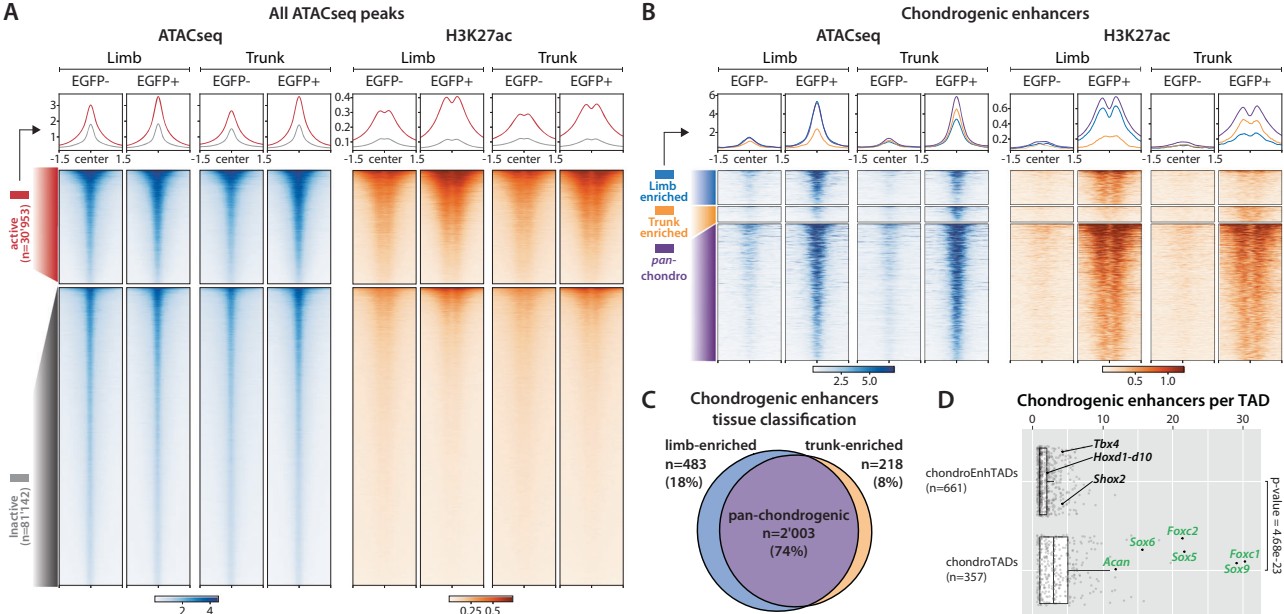

**Fig. 2 | Mapping and characterizing chondrogenic enhancers in limbs and trunk. A** Distribution of ATAC-seq and H3K27ac ChIP-seq signal over all accessible regions in EGFP+ and EGFP- cells from limbs and trunks. Two categories of regions are defined: accessible regions without H3K27ac enrichment ("*inactive*") and accessible regions with H3K27ac enrichment in one of the studied conditions ("*active*"). **B** Distribution of ATAC-seq and H3K27ac ChIP-seq signal over the 2'704 chondrogenic enhancers identified in this work: accessible regions with differential H3K27ac coverage between EGFP+ and EGFP- cells (≥ 4-fold change coverage). **C** Tissue specificity of the 2'704 chondrogenic enhancers according to limb- and trunk-enrichment. Enrichment in limb or trunk is defined by difference in H3K27ac coverage ≥2-fold change in comparison to the counter-part tissue. **D** Boxplots showing the distribution of number of chondrogenic enhancers per TAD, split by two different TAD categories: TAD with chondrogenic enhancers and at least one

protein-coding chondrogenic gene (*chondroTADs* n = 357, chondrogenic enhancers n = 1'363 (median=3) and chondrogenic genes *n* = 478) and TAD with chondrogenic enhancers but without any chondrogenic-specific gene (*chondroEnhTADs n* = 661, chondrogenic enhancers *n* = 1'315 (median=1)). Dots represent the number of chondrogenic enhancers contained in each TAD and some relevant examples are named based on the genes they encompass. Total number of TAD is 3'109, extracted from E14.5 mouse forelimb cartilage Hi-C[29]. Boxes indicate the first and third quartiles, the whiskers indicate ±1.5 × interquartile range and the horizontal line within the boxes indicates the median. Statistical test used in D: Two-tailed Wilcoxon rank sum test, with a p-value of 4.68e[-23]. ATAC-seq and H3K27ac coverages over 3 kb are centered at the corresponding merged ATAC-seq peaks located within a 75 bp window.

development (Supplementary Data 13). In contrast, protein-coding genes in *chondroEnhTADs* (n = 4'723, Supplementary Data 14) were enriched for general developmental and morphogenetic processes (Supplementary Data 15) and included known patterning transcription factors genes such as *Shox2*, *Hoxd* and *Tbx4*. This suggests that chondrocyte enhancers in *chondroEnhTADs* may control the chondrogenic expression of more general developmental pathways that are not exclusive to chondrocytes but maintained during chondrogenic differentiation.

Next, we examined if the trunk- or limb-specific expression of chondrogenic genes in *chondroTADs* correlates with limb- or trunk-enriched activity of chondrogenic enhancers in the same *chondroTADs*. We selected 337 *chondroTADs* containing either exclusively limb-, trunk-specific or *pan*-chondrogenic genes as defined in Fig. 1H. We categorized the enhancer content within each TAD: limb, trunk, *pan*-chondrogenic or a combination thereof (Supplementary Fig. 6). *Pan*-chondrogenic expressed genes were generally (> 95% of cases) associated with *pan*-chondrogenic enhancers or a mix of different enhancer types. In contrast, limb- or trunk-expressed chondrogenic genes were commonly found in TADs with corresponding limb- or trunk-enriched chondrogenic enhancers (for example 8/17 for limb expressed *chondroTADs*). These results support that limb- and trunk-enriched chondrogenic enhancers in *chondroTADs* commonly drive the limb- and trunk-specific expression of protein-coding chondrogenic genes. However, cases of limb- or trunk-restricted genes co-located with *pan*-chondrogenic enhancers suggest that additional regulatory mechanisms may be involved in conferring additional tissue specificity, e.g., through tissue-specific changes in chromatin 3D structure[30].

## Binding of transcription factors at chondrogenic enhancers

As early chondrocytes start to differentiate in the limb bud between E10.5 and E11.5[4], we set out to describe the dynamics of activation of chondrogenic enhancers over developmental time. To do so, we used available ATAC-seq dataset from different bulk limb samples ranging from E9.75 to E14.5[31,32]. We observed that most chondrogenic enhancers become accessible between E9.75 and E11.5 and remain open until E14.5 or beyond (Supplementary Fig. 7). This suggests that many of the active chondrogenic enhancers we observed at E14.5 are already active or poised at earlier developmental stages. Moreover, it also suggests that the expression of transcription factors that enable these set of enhancers to become accessible must be initiated between E9.75 and E11.5.

We then investigated which transcription factors (TFs) might bind to and control the activities of chondrogenic enhancers. We examined the 2'704 candidate chondrogenic enhancer regions for the presence of 356 different TF binding sites (TFBSs) and determined their possible enrichment relative to accessible chromatin regions that showed no H3K27ac signal in chondrogenesis (Fig. 2A)[33]. Several TFBSs were depleted in chondrogenic enhancers, including binding sites for the architectural proteins CTCF and MAZ, as well as for various non-chondrogenic factors including the muscle related MYOD1, MYOG or MYF6 (Fig. 3A, Supplementary Fig. 8A). In contrast, we observed enrichment for SOX9, MEF2C/D, and several FOX factors binding sites in putative chondrogenic enhancers. These findings are consistent with the preferential expression of *Sox9*, as well as *Foxc1 and Foxc2* in EGFP+ cells (Supplementary Data 3), their transcription onset between E9.5 and E11.5[34,35], as well as the known involvement of these factors in chondrogenesis[36–39]. We then measured whether limb- and trunk-

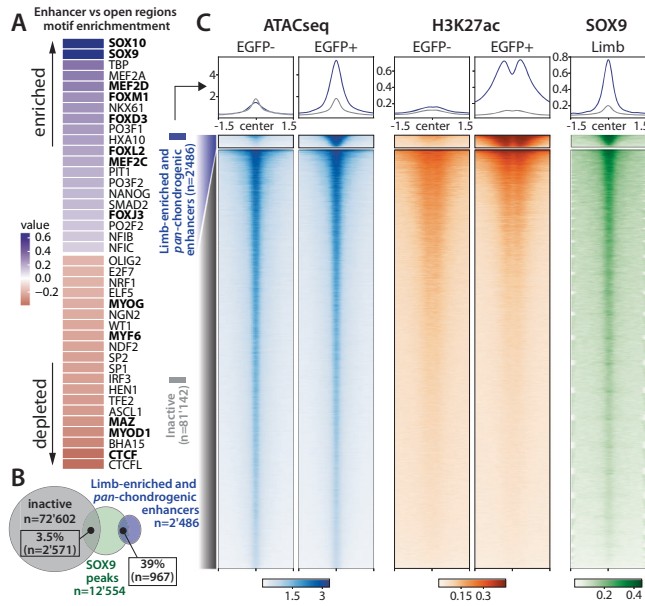

**Fig. 3 | Transcription factor binding analysis. A** Log$_2$ fold-change of motif enrichment between chondrogenic enhancers and "*inactive*" accessible regions for the top 20 motifs enriched and depleted. **B** Overlap of SOX9 binding peaks[40] with limb-enriched and *pan*-chondrogenic enhancers (*n* = 2'486) compared to "*inactive*" accessible regions (see Fig. 2A). For better comparison between enhancers and inactive regions, the latter were merged when closer than 500 bp (*n* = 72'602). **C** Binding densities of SOX9[40] at limb-enriched and *pan*-chondrogenic enhancers compared to inactive accessible regions (see Fig. 2A). Heatmaps are generated from the limb EGFP+ and EGFP- datasets. ATAC-seq, H3K27ac and SOX9 coverages over 3 kb centered at the corresponding merged ATAC-seq peaks located within a 75 bp window.

enriched enhancers displayed differential transcription factor motifs and only observed a limited enrichment of PRXX2 motif in the limb-enriched enhancers compared to the trunk-enriched ones (Supplementary Fig. 8B).

Considering the known role of SOX9 in chondrocyte development[1], SOX9 binding has been used to identify initial sets of SOX9-dependent candidate enhancers in postnatal rib development[9]. To examine to what extent SOX9 is commonly binding to chondrogenic enhancers in vivo during critical early stages of skeletal development, we intersected our genome-wide chondrocyte enhancer set with SOX9 chromatin binding sites observed by ChIP-seq in E13.5 murine limbs (Supplementary Data 16)[40]. We observed that 967 (39%) putative limb chondrogenic enhancers overlapped with in vivo SOX9 binding sites in E13.5 limbs (Fig. 3B). We did not observe a differential proportion of SOX9-bound enhancers in *chondroTADs* (38.7%, 498/1'287) and *chondroEnhTADs* (39%, 458/1'175). In contrast, only 3.5% of accessible chromatin regions without H3K27ac signal ("inactive" regions in Fig. 2A) were bound by SOX9. This observation was further supported through comparisons of local SOX9 binding density maps of chondrogenic enhancers with accessible chromatin regions that showed no H3K27ac signal or non-chondrogenic enhancers (Fig. 3C, Supplementary Fig. 8C). However, our study also highlights the limitations of using a single central transcription factor for enhancer discovery. We indeed observed 1'519 additional regions (61% of enhancers) with limb-enriched or *pan*-chondrogenic enhancer signatures that did not overlap with called limb SOX9 binding sites[40]. Together, these results indicate that the approach used here provides a comprehensive profile of SOX9-dependent and -independent enhancers active in developing chondrocytes.

## Variability in stature is associated to multiple enhancer-rich chondrogenic loci

Developmental control of chondrogenesis can modulate several features of skeletal growth. One of the most studied of these outcomes is stature and its variability across populations[3,41]. A search for loci associated with tibial length using selective breeding of outbred mice over multiple generations identified eight regions significantly contributing to the selected trait[42]. Across these eight regions, we identified 15 chondrogenic genes, including *Sox9* (Fig. 4A, Supplementary Fig. 9A). Looking at the distribution of enhancers across these regions, we only found 5 non-chondrogenic enhancers (0.6% of all non-chondrogenic enhancers) but 73 chondrogenic enhancers (2.7% of all chondrogenic enhancers), suggesting a potential contribution of these enhancers to the increased tibial length observed in the Longshanks mice (Fig. 4A). In fact, at the *Nkx3-2* locus, a locus known to be involved in short stature in human[43], a single nucleotide variation in previously described enhancer N1 is associated with changes in tibial length[42]. Consistent with these observations, the N1 element is identified as a chondrogenic enhancer. Moreover, we also discovered two additional chondrogenic enhancers at the locus, including a second SNP-containing region (N2) already identified by Castro et al[42]. and a chondrogenic enhancer (CE1) (Supplementary Fig. 9B).

In humans, a recent genome-wide association study of 5.4 M individuals resulted in a saturated map of common variants associated with adult stature and a set of 7'209 height variance-explaining loci[44]. Among height variance-explaining loci, we noted well-known chondrogenic genes that collectively bear hundreds of putative chondrogenic enhancers. This includes the *ACAN* gene, that alone accounts for 0.24% of height variance, as well as *GDF5*, *SOX9*, *FGFR3*, and *COL2A1*[3,44]. We thus decided to test whether chondrogenic enhancers could provide a mechanistic framework to understand the genetic variation at these loci. To do so, we computed the variance explained by the overlap between the 6'916 mouse-conserved height variance-explaining loci (mHVEL) accounting for 42.5% of height variance with chondrogenic enhancers. We observed that chondrogenic enhancers overlapped 794 (11.5%) of them and could provide an interpretation framework for 24.4% of mHVEL variance (Supplementary Data 17). Yet, as most height variance-explaining loci contain both coding and non-coding segments, the variance explained by the overlap of mHVEL with chondrogenic enhancer is also accounting for variations within coding parts of genes, and particularly chondrogenic genes. Therefore, to focus on variation occurring at non-coding segments only we decided to further focus our analysis on the 1'771 mHVEL deprived of any protein-coding gene (non-coding mHVEL). Cumulatively, non-coding mHVEL explain 5.7% of the height variance while the 169 (9.5%) of them overlapping with chondrogenic enhancers account for 18.4% of this variance (Supplementary Data 17). We then aimed at measuring whether chondrogenic enhancer are more likely to explain height variance than other enhancer regions.

To do so, we measured the overlap of mHVEL and non-coding mHVEL with three enhancer categories: chondrogenic enhancers in *chondroTADs*, in *chondroEnhTADs* as well as non-chondrogenic enhancers. To normalize for the size of each enhancer category, we selected 877 enhancers displaying the highest differential coverage of H3K27ac between EGFP- and EGFP+ cells and located on autosomes. Looking at mHVEL, we found that chondrogenic enhancers from *chondroTADs* overlapped with greater height variance-explaining loci than the two other enhancer categories (Supplementary Fig. 10). Looking at non-coding mHVEL, we found that the overlap of chondrogenic enhancers from both *chondroEnhTADs* and *chondroTADs* explained a higher cumulative variance than the one with non-chondrogenic enhancers (Fig. 4B). We also noted that *chondroEnhTADs* enhancers had a higher variance explaining overlap than *chondroTADs* ones. This suggests that height variants affecting chondrogenic enhancers, which control general developmental

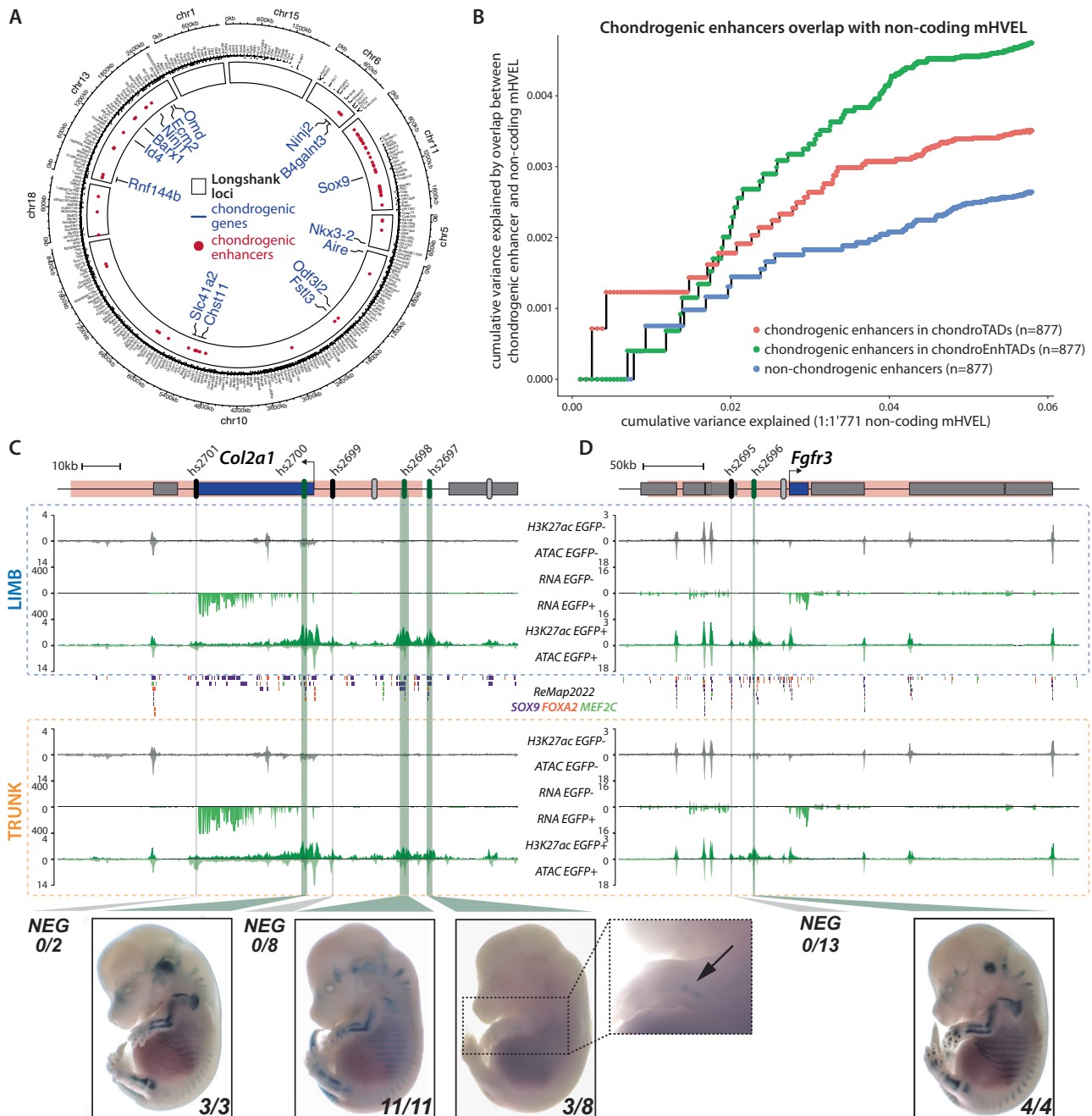

**Fig. 4 | Variability in stature is associated to chondrogenic enhancers.**
**A** Distribution of chondrogenic enhancers (red circles) and genes (blue names) in eight loci associated to the *Longshank* increased tibia size[42]. On the outside are displayed other protein coding genes and relative coordinates. **B** Cumulative height variance explained by the 1'771 non-coding mHVEL from[44] (x-axis, regions sorted from highest to lowest variance) and the same variance explained by the enhancers overlapping the mHVEL (y-axis). Note that chondrogenic enhancers explain more variance than non-chondrogenic enhancers. Enhancers on the X and Y chromosome were excluded from the analysis. **C**, **D** RNA-seq, ATAC-seq and H3K27ac ChIP-seq normalized coverages in limb (blue box) and trunk (orange box) EGFP+ and EGFP- cells at the *Col2a1* and *Fgfr3* loci. mHVEL are highlighted in pink[44].

Predicted active, inactive and untested enhancers are shown by green, black and grey ovals, respectively. Green and grey vertical bars highlight tested active and inactive regions. Binding events for SOX9, FOX2A and MEF2C are displayed in-between limb and trunk boxes[56]. EGFP+ datasets are colored in green, EGFP-datasets in grey. RNA-seq and ATAC-seq coverages are an average of two replicates. **C** 7 enhancers are predicted at the *Col2a1* locus, 5 were tested in an in vivo reporter assay. Hs2700 (also E2 enhancer from ref. [111]), hs2698 and hs2697 displayed a chondrogenic pattern. **D** 3 enhancers are predicted at the *Fgfr3* locus, 2 were tested in an in vivo reporter assay and hs2696 displayed a chondrogenic pattern. Protein-coding genes are represented as grey or blue boxes.

genes, may be equally or even more functionally significant than those controlling chondrogenic genes. In summary, these results highlight the potential of chondrogenic enhancers as a mechanistic framework to interpret how non-coding variation can influence adult height.

## Human height loci show complex landscapes of chondrogenic in vivo enhancers

As the activities of chondrogenic loci are linked to adult height, we tested if our dataset can be used to elucidate the enhancer landscape of these loci. First, we examined if previously described enhancers at

these loci are correctly re-identified by our data. At *Acan*, we predict 11 enhancers of which 5 were previously shown to have chondrogenic activity (Supplementary Fig. 11A)[45,46]. At the *Gdf5* locus, we predict 4 enhancers including 3 with known chondrogenic activity in vivo (Supplementary Fig. 11B)[47,48]. Similarly, at *Sox9*, we predict 27 chondrogenic enhancers, 4 of which have validated in vivo activity in chondrocytes (Supplementary Fig. 11C)[6,49,50].

To test the accuracy of our enhancer set prediction, we selected the *Col2a1* and *Fgfr3* chondrogenic loci that are both associated with height variation, skeletal defects and short/tall statures in mouse and human[3,44,51–54]. Here, we predict 7 and 3 chondrogenic enhancers at *Col2a1* and *Fgfr3*, respectively (Fig. 4C, D). We cloned the human sequence of 7 of these putative enhancers into a LacZ reporter vector and used site-directed integration[55] to test their activity in vivo at embryonic stage E14.5 with minimal position effect. At the *Col2a1* locus, 3 out of 5 tested enhancers showed skeletal staining, while 2 showed no reproducible activity (Fig. 4C). Of note, 2 of the active enhancers (hs2697, hs2698) were also found active in neonatal rib chondrocytes suggesting a stability of the enhancer activity over time[9]. The 2 inactive enhancers displayed fewer SOX9, FOXA2 and MEF2C binding events according to the *remap032022-mm39* database than the 3 active ones (Fig. 4C)[56]. At the *Fgfr3* locus, 1 out of 2 tested enhancers showed skeletal activity while the other one did not display any activity (Fig. 4D). In total, 4 predicted enhancers displayed chondrogenic in vivo activity while 3 candidate sequences did not produce any staining, highlighting the specificity of the predictive approach.

## Functional assessment of four predicted chondrogenic enhancer regions

The capacity of enhancers to drive reporter genes shows their potential to control transcription in a specific tissue, yet, it does not show if and how they act on endogenous target genes in vivo. It is therefore essential to functionally validate whether identified enhancers control the transcription of predicted associated genes. To do so, we selected four loci pertinent to skeletal development and stature and produced homozygous chondrogenic enhancer deletions in the *Col2a1^EGFP* sensor background. We then investigated by bulk RNA-seq whether these deletions affected the expression of their predicted target genes in the limbs and trunk of mouse E14.5 fetuses.

We first furthered our investigation of the validated *Col2a1* and *Fgfr3* loci enhancers (Fig. 4C, D). At the *Col2a1* locus we engineered a homozygous 9.1 kb deletion of two validated *pan*-chondrogenic enhancers, hs2697 and hs2698 (*Col2a1^EGFP;Δhs2697-2698*, Fig. 4C, Fig. 5A). RNA-seq analyses of *Col2a1^EGFP;Δhs2697-2698* E14.5 fetuses revealed a significant 20% reduction in *Col2a1* expression in the trunk but not in the limbs (Fig. 5B, Supplementary Data 18). Notably, EGFP transcript levels and fluorescent signal also exhibited a substantial decrease in the trunk samples, implying that the sensor cassette was also affected by the deletion (Supplementary Fig. 12A, B). Similarly, at the *Fgfr3* locus, we generated a homozygous 5.9 kb deletion removing the hs2696 chondrogenic enhancer (*Col2a1^EGFP;Fgfr3^Δhs2696*, Fig. 4D, Fig. 5C). Remarkably, we observed that the deletion of this single enhancer was sufficient to reduce *Fgfr3* expression by as much as 75% in limbs and trunk of E14.5 *Col2a1^EGFP;Fgfr3^Δhs2696* fetuses (Fig. 5D, Supplementary Data 18).

We next examined two enhancer regions overlapping with non-coding mHVEL at the *Hhip* and *Nkx3-2* loci. At the *Hhip* locus, a gene linked to impaired skeletal mineralization in mice[57], we deleted two limb-enriched chondrogenic enhancers (CE2-CE3) identified within the most confident non-coding mHVEL (*Col2a1^EGFP;Hhip^ΔCE2-3*, Fig. 5E, homologous region of METAFE:2053 from[44]). We found that the deletion of these limb-enriched chondrogenic enhancers resulted specifically in a limb, but not in a trunk, reduction of *Hhip* transcripts by 15% in E14.5 *Col2a1^EGFP;Hhip^ΔCE2-3* fetuses (Fig. 5F, Supplementary

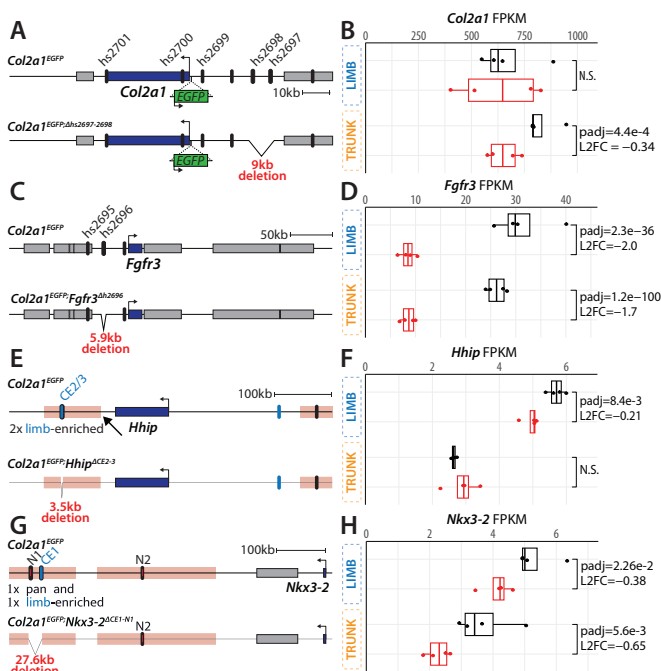

**Fig. 5 | Functional assessment of putative chondrogenic enhancer in vivo.** **A** Schematic representation of the homozygous 9 kb deletion of hs2697 and hs2698 *Col2a1* chondrogenic enhancers. **B** Boxplots showing the distribution *Col2a1* FPKMs obtained from control (black) or *Col2a1^EGFP;Δhs2697-2698* (red) E14.5 limbs and trunk. **C** Schematic representation of the homozygous 5.9 kb deletion of hs2696 *Fgfr3* chondrogenic enhancer. **D** Boxplots showing the distribution of *Fgfr3* FPKMs obtained from control (black) or *Col2a1^EGFP;Fgfr3^Δhs2696* (red) E14.5 limbs and trunk. **E** Schematic representation of the homozygous 3.5 kb deletion of 2 limb-enriched (CE2-3) *Hhip* chondrogenic enhancer. Non-coding mHVEL are depicted in pink (arrow highlights the mHVEL homologous to METAFE:2053, the highest non-coding mHVEL)[44]. **F** Boxplots showing the distribution of *Hhip* FPKMs obtained from control (black) or *Col2a1^EGFP;Hhip^ΔCE2-3* (red) E14.5 limbs and trunk. **G** Schematic representation of the homozygous 27.6 kb deletion of 2 *Nkx3-2* chondrogenic enhancer (N1 and CE1). Non-coding mHVEL are depicted in pink. Chondrogenic N1 and N2 enhancers from ref. 42 are highlighted. **H** Boxplots showing the distribution of *Nkx3-2* FPKMs obtained from control (black) or *Col2a1^EGFP;Nkx3-2^ΔCE1-N1* (red) E14.5 limbs and trunk. Limb-enriched, trunk-enriched, and *pan*-chondrogenic enhancers are represented respectively as blue, orange, and black ovals. Protein-coding genes are represented as grey or blue boxes. In boxplots, boxes indicate the first and third quartiles, the whiskers indicate ±1.5 × interquartile range, and the horizontal line within the boxes indicates the median. Statistical test used: DESeq2 Wald test, padj is the FDR-corrected by Benjamini-Hochberg method two-tailed *p*-value, and L2FC is the log₂ fold change estimated by DESeq2 on the gene raw counts. $N = 4$ biologically independent samples per condition. Source data are provided as a Source Data file.

Data 18). At the *Nkx3-2* loci we deleted a 27.6 kb region encompassing a limb-enriched (CE3) and the N1 *pan*-chondrogenic enhancer previously associated with longer tibiae in Longshank mice (*Col2a1^EGFP;Nkx3-2^ΔCE1-N1*, Fig. 5G, Supplementary Fig. 9B)[42]. Remarkably, this deletion resulted in a 25% and 37% *Nkx3-2* expression loss in limbs and trunk, respectively, of E14.5 *Col2a1^EGFP;Nkx3-2^ΔCE1-N1* fetuses (Fig. 5H, Supplementary Data 18). In addition to the observed reduction of *Nkx3-2* transcripts, we also observed major changes in downstream chondrogenic genes, including *Sox9*, *Col2a1*, and the EGFP reporter (Supplementary Fig. 12C). This last observation was also directly supported by a clear reduction of the distribution of the GFP signal as quantified by flow cytometry in both limbs and trunk of E14.5 *Col2a1^EGFP;Nkx3-2^ΔCE1-N1* fetuses (Supplementary Fig. 12D).

In summary, functional evidence at these four loci illustrates how our genome-wide set of predicted chondrogenic enhancers can be used to assess the regulatory landscape of individual loci, thereby

providing a framework for linking human variation to gene regulatory activity.

## Discussion

Because of the cellular heterogeneity of developing tissues, cell type-specific enhancers tend to be poorly characterized in vivo. Here, we used a *Col2a1* regulatory sensor to mark pre-hypertrophic chondrocytes combined with cell sorting and chromatin profiling. In contrast to a previously described *Col2a1* transgene-based approach[58–60], this EGFP knock-in bears all the regulatory specificities of the *Col2a1* locus. This is particularly visible as the loss of two *Col2a1* distal enhancers, that are not included in the transgene-based approach, showed a significant reduction of *Col2a1* and EGFP expression in the trunk. Generally, this result showcases the importance of a complete regulatory landscape for driving accurate reporter gene expression.

Using stringent thresholds, we identified 2'704 pre-hypertrophic chondrogenic enhancers based on chromatin accessibility and H3K27ac signal in chondrogenic cells compared to non-chondrogenic ones. The usage of H3K27 acetylation to define active enhancers in our dataset explains the reduced number of high-confidence enhancers compared to other studies based solely on chromatin accessibility[11,12,28]. Generally, we observed that chondrogenic enhancers become accessible between E9.75 and E11.5 at a time where *Sox9*, *Foxc1*, and *Foxc2* start their expression, suggesting they might play a role in establishing open chromatin at these regions. In fact, SOX9 as well as several FOX genes transcription factors have been shown to act as pioneer factors in different biological context and could therefore indeed act early on to open and prime chondrogenic enhancers for activity[61,62].

A subset of enhancers defined in this study were functionally validated by both in vivo reporter assay and targeted deletions. On top of providing confidence in the accuracy of the present dataset, these results pinpoint the potential effect of non-coding mutations leading to loss-of-function skeletal phenotypes. At the *Col2a1* locus, the loss of 2 out of 7 chondrogenic enhancers resulted in a reduction of *Col2a1* transcript in the trunk and could therefore lead to defects in endochondral ossification as seen in the gene loss-of-function experiment[51]. As the deletion of the hs2696 enhancer resulted in a dramatic decrease of *Fgfr3* transcript, variants affecting enhancers could induce Camptodactyly, Tall Stature, and Hearing Loss (CATSHL) syndrome as known for the *Fgfr3* loss-of-function in humans[53,54]. Finally, the deletion of 2 out of the 3 chondrogenic enhancers located at *Nkx3-2* locus resulted in a strong reduction of *Nkx3-2* transcripts and global reduction in the expression level of several key chondrogenic genes as diverse as *Sox9*, *Col2a1* and *Ihh*. Variation in these enhancers could therefore result in *Nkx3-2*-mediated skeletal defects *as seen* in mice[63] and humans[43].

We observed frequent colocalization of chondrogenic enhancers with chondrogenic genes, with 1'363 enhancers embedded in 357 TADs that also contained 478 protein-coding chondrogenic genes. This observation supports a function for these enhancers in controlling chondrogenic differentiation and growth. In most cases, enhancers and their target genes were active in both limb and trunk chondrocytes. Yet, in roughly 10% of the cases, we observed chondrogenic genes preferentially expressed in one of the two tissues. These genes were then found regulated both by limb- or trunk- enriched enhancers and by *pan*-chondrogenic enhancers. While the concept of limb enhancers regulating limb-specific genes and trunk enhancers regulating trunk-specific genes seems coherent, the idea of *pan*-chondrogenic enhancers regulating limb- or trunk-specific chondrogenic genes is counterintuitive. In these cases, the tissue-specific activity of *pan*-chondrogenic enhancers could be further modulated through additional regulatory mechanisms, such as tissue-specific changes to the three-dimensional local chromatin topology. Such a scenario has recently been demonstrated at the *Pitx1* locus, where an enhancer

active in both fore- and hindlimb is spatially sequestered away from the repressed gene in forelimbs, restricting the developmental activity of the *pan*-limb-specific enhancers to the developing hindlimb[30].

We also observed 1'315 chondrogenic enhancers spread over 661 TADs containing 4'723 protein-coding genes that do not seem to show chondrocyte-specific expression but include numerous genes with known functions in spatiotemporal patterning of organs and embryonic structures. Examples include the *Hoxd* gene cluster, for which expression is initiated in undifferentiated mesenchyme but maintained in chondrocytes[64]. Here, our data identified a single element with a limb chondrocyte-specific enhancer signature, CS93, at a location that coincided with a known chondrogenic limb enhancer[65]. At the *Shox2* locus, three chondrogenic enhancers mirrored *Shox2* transcriptional activity, with active enhancer signatures observed in limb chondrocytes but not in trunk chondrocytes, where *Shox2* is lowly transcribed. In fact, *Shox2* expression in differentiating proximal limb chondrocytes was shown to be vital for bone growth[66,67]. This finding suggests that the identified chondrogenic enhancers play a crucial role in controlling *Shox2* expression during bone growth. Together, these examples show that at late developmental stages patterning loci might rely on tissue-specific enhancers, such as chondrogenic ones, to maintain their activities.

Most chondrocyte-specific enhancers discovered in this study have orthologous conserved sequences in the human genome (2'463 of the 2'704, Supplementary Data 19), supporting the value of this data for the interpretation of non-coding variants observed in human patients with genetically unexplained rare skeletal diseases. To illustrate this approach, we examined the chondrogenic enhancer landscape at the *SOX9* locus where deleterious mutations in the *SOX9* gene itself have been shown to cause campomelic dysplasia (CD), a severe bone malformation[68,69]. Recently, a case of acampomelic CD, which is an atypical type of CD without the eponymous bowed limb phenotype, was shown not to carry a coding pathogenic variant in the SOX9 gene[70]. While many campomelic dysplasia patients do not survive beyond infancy, this individual had reached the age of 34-year-old at the time of the publication. The patient had a translocation upstream of *SOX9* that relocates 15 out of the 26 predicted SOX9 chondrogenic enhancers from chromosome 17 to chromosome 11 (Supplementary Fig. 11C). This loss of more than half, but not all chondrogenic *SOX9* enhancers offers a plausible mechanistic explanation for the relatively mild form of the disease and the survival of the individual.

In mice and in humans, chondrogenic enhancers can be used as a mechanistic link between non-coding genetic variation and changes in adult height and stature. Here, we showed that between 18% (considering only non-coding mHVEL) to 24% (considering all mHVEL) of the height variance explained overlapped chondrogenic enhancers, while the rest of the variance could be associated either to regions involved in other parts of bone formation and to regions linked to height variance but unrelated to chondrogenesis. Genome-wide association studies have had great success in identifying common variants associated with common bone-related phenotypes including adult stature, osteoporosis and osteoarthritis[71]. However, aside from limited examples[12], these studies have had much less success in mapping non-coding variant associations to causal genes and regulatory elements. As whole genome sequencing is becoming more accessible, the full enhancer dataset presented here and the validated enhancers will offer a panel that could greatly help in the process of identifying causal genetic variations.

## Methods
### Animal procedures
Animal work performed in Geneva adheres to all relevant ethical regulations of the University of Geneva and follows procedures approved by the animal care and experimentation authorities of the Canton of Geneva, Switzerland (animal protocol numbers GE/89/19 and GE192A).

Animal work performed at Lawrence Berkeley National Laboratory (LBNL) was reviewed and approved by the LBNL Animal Welfare Committee under protocol numbers #290003 and #290008.

## Statistics and reproducibility
No statistical method was used to predetermine sample size. No data were excluded from the analyses. The experiments were not randomized, and the investigators were not blinded to allocation during experiments and outcome assessment.

## CRISPR/Cas9 engineered alleles
Genetically engineered alleles were generated using the CRISPR/Cas9 editing according to ref. 72. Briefly, sgRNA was designed using the online software Benchling (https://benchling.com) and was chosen based on predicted off-target and on-target scores. *Col2a1* sgRNA was cloned in the pX459 plasmid (Addgene, 48139) and designed to target CRISPR–Cas9 to chr15:97903847 (mm39). To construct the homozygous *Col2a1^EGFP^* mESC clone, the EGFP sensor from[73] was adapted by exchanging homology arms. mESCs culture and genetic editing followed standard procedure[72]. Briefly, G4 mouse embryonic stem cells were transfected with 4 μg of EGFP–cassette and 8 μg of pX459 vector containing the sgRNA. *Col2a1^EGFP^* mESC clones can be obtained upon request. The chondrogenic enhancers deletion alleles were produced from the same *Col2a1^EGFP^* mESC clone, co-transfected with 8 μg of pX459 vector containing each one sgRNA. All the genotyping primers and sgRNA sequences can be found in Supplementary Data 20.

## Aggregation of mESC
Fetuses were generated by tetraploid complementation from G4 male mESCs obtained from the Nagy laboratory (http://research.lunenfeld. ca/nagy/?page=mouse%20ES%20cells)[74,75]. Desired mESCs were thawed, seeded on male and female CD1 feeders and grown for 2 days before the aggregation procedure. Donor male and female tetraploid embryos were provided from in vitro fertilization using C57Bl6J x B6D2F1 backgrounds. Aggregated embryos were transferred into CD1 foster females. Because they derive from G4 mESCs, derived fetuses were male. All animals were obtained from Janvier laboratories.

## Light sheet microscopy imaging
An E14.5 fetus was post-fixed overnight in 4% PFA and processed for light-sheet imaging as described in[73]. Briefly, tissue was cleared using passive CLARITY based clearing method. Briefly, tissue was incubated in a Bis-free Hydrogel X-CLARITY™ Hydrogel Solution Kit (C1310X, Logos Biosystems) for 3 days at 4 °C, allowing diffusion of the hydrogel solution into the tissue. Polymerization of solution was carried in a Logos Polymerization (C20001, Logos Biosystem) system at 37 °C for 3 h. After two washes of 30′ in PBS, samples were immersed in a SDS based clearing solution (SDS-Clearing solution: for 2 L of 4% SDS solution used 24.73 g of boric acid (Sigma B7660 or Thermofisher B3750), 80 g of sodium dodecyl sulfate (Brunschwig 45900-0010, Acros 419530010 or Sigma L3771), in dH₂O, final solution pH 8.5) and left at 37 °C for 48 h. Once cleared, tissue was washed twice in PBS-TritonX 0.1% and then placed in a Histodenz based-refractive index-matching solution (Histodenz Sigma D22158, PB + Tween + NaN₃ pH 7.5 solution, 0.1% Tween-20, 0.01% NaN₃, in 0.02 M phosphate buffer, final solution pH 7.5). Imaging was performed with a home-built mesoscale single-plane illumination microscope; complete description of the mesoSPIM microscope is available in[76]. Briefly, using one of the two excitation paths, the sample was excited with 488 and 647 nm laser. The beam waist was scanned using electrically tunable lenses (ETL, Optotune EL-16-40-5D-TC-L) synchronized with the rolling shutter of the sCMOS camera. This produced a uniform axial resolution across the field-of-view (FOV) of 5 μm. Signals were filtered with 530/43 nm and LP663 longpass filters (BrightLine HC, AHF). Z-stacks were acquired at 5 μm spacing with a zoom set at ×1 resulting in an in-plane pixel size of 6.55 μm. Images were pre-processed to subtract the background and autofluorescence signal using the 647 nm excitation channel and subsequent normalization and filtering of the images were performed with *Amira* v2021.1. 3D video and images were captured using *Imaris* v9.8.0 software.

## Genomic data
The NGS datasets presented here were mapped either on GRCm39/mm39[77] or on a customized GRCm39/mm39_eGFP-SV40pA genome. The GTF annotations used in this work derive from ENSEMBL GRCm39 release 104[77] and are filtered against read-through/overlapping transcripts, keeping only transcripts annotated as 'protein-coding' for 'protein-coding' genes, thus discarding transcripts flagged as 'retained_intron', 'nonsense-mediated decay' etc., to conserve only non-ambiguous exons and avoid quantitative bias during data analysis by STAR/Cufflinks[78]. GRCm39/mm39_eGFP-SV40pA sequence, the filtered GTF file and the scripts used to construct it are available on Zenodo (https://doi.org/10.5281/zenodo.7837435). All NGS libraries were sequenced by the iGE3 Genomics platform (www.ige3.genomics. unige.ch).

## Tissue collection and cell preparation
Tissues were prepared following a similar protocol for flow-cytometry analysis, FACS-sorting for RNA-seq, ChIP-seq and ATAC-seq and single-cell RNA-seq. Briefly, limbs and trunk samples were dissected from decapitated E14.5 fetuses in cold PBS solution. Tissues were first minced using a pair of micro-scissors. After PBS removal, a single cell suspension was achieved by incubating tissues in 1.2 mL Trypsin-EDTA (Thermo Fischer Scientific, 25300062) for 15′ to 18′ at 37 °C in a thermomixer with resuspension steps each 6′. After blocking with one volume of 5% BSA (Sigma Aldrich, A7906-100G), cells were passed through a 40 μm cell strainer for further tissue disruption and another volume of 5% BSA was added to the cell strainer to pass leftover cells. Cells were then centrifuged at $400 \times g$ for 5′ at 4 °C and, after discarding the supernatant, they were resuspended in 1% BSA for cell sorting. 5 mM of Na-Butyrate were added to the BSA when planning for subsequent fixation for H3K27Ac-ChIP.

## Single-Cell RNA-seq processing
Cells from a single-cell suspension of dissociated *Col2a1^EGFP^* E14.5 limbs were counted and processed on a Chromium Single Cell 3' GEM, Library & Gel Bead Kit v3 following the manufacture's protocol (10X Genomics, PN-1000075). Library was then sequenced on an Illumina HiSeq 4000 with paired-end reads of 28/90 bases.

## scRNA-seq analysis
Reads were mapped to a customized GRCm39/mm39_eGFP-SV40pA reference genome and corresponding gene annotation as for the bulk RNA-seq using 10X Genomics *Cell Ranger* v6.1.2[79] and data analyzed with the R package *Seurat* v4.3.0[80]. Briefly, *Cell Ranger* filtered_feature_bc_matrix.h5 matrix was first imported into *Seurat* (min.cells = 3, min.features = 200), filtered (nFeature_RNA > 200 & nFeature_RNA < 5000 & percent.mt < 5 & nCount_RNA > 1000 & nCount_RNA < 26000), log-normalized and scaled. The reporter gene *eGFP-SV40pA* was excluded from the list of variable gene in all subsequent normalizations. Scaled data were then used for principal component analysis (PCA) with npcs=100 and non-linear dimensional reduction by Uniform Manifold Approximation Projection (UMAP)[81] ndims=1:100 as input. We then identified and excluded doublets using the R package *DoubletFinder* v2.0.3[82] (PCs = 1:100, pN = 0.25, pK =0.07, nExp =55, reuse.pANN = FALSE, sct = FALSE). Cells were then further filtered to exclude blood cell present in our dataset (percent.mt > 1 & percent.mt <5). We then applied a first SCTransform normalization[83] on our dataset, scored the cell-cycle and performed a second SCTransform normalization to regress it out. Following this regression, cells were

then clustered using PCA (npcs = 100), UMAP (dims = 1:100) and nearest neighbors of each cell were calculated (dims=1:100). Clusters were determined using *Seurat FindClusters* function with default parameters and a resolution of 1.1. In that way 16 clusters were defined (n = 2041 cells). Identification of clusters identity was done by using *Seurat FindAllMarkers* on the RNA assay and mesenchyme clusters were then merged. The list of marker genes is provided in Supplementary Data 1. Since the interest of this work was focus on the mesenchymal populations of cells that express *Col2a1*, we then subsetted and re-clustered the mesenchyme cluster alone (n = 1617 cells). To do so we repeated for the subsetted cells a SCTransform normalization regressing out the cell-cycle and perfomed PCA with npcs=50. UMAP embedding was calculated with ndims=1:50 and cluster resolution was set at 0.8 after finding neighbors with dims=1:50 to reveal subpopulations. We observed 8 mesenchyme subpopulations that we named according to their marker genes. Identity markers were found using *FindAllMarkers* on the RNA assay and are provided in Supplementary Data 2. DotPlots and FeaturesPlots were generated from the RNA assay of the Seurat objects. To correct the distribution of expression from sampling noise, we used *baredSC* v1.0.0[23] (--minNeff 200 --xmax 6). This allows us to evaluate a 68% confidence interval on the expression distribution per cluster. *BaredSC* was also used to compute the co-expression distribution per cluster in Supplementary Fig. 2B where correlation was given with 68% confidence interval and p-value indicated is the mean probability + estimated standard deviation on the mean probability. *BaredSC* scripts are available at https://doi.org/10.5281/zenodo.10806675. The joint density UMAP was produced using the R packages *Nebulosa* v1.6.0[84] and *scCustomize* v1.1.0 (https://doi.org/10.5281/zenodo.5706430).

### Cell sorting

Several E14.5 *Col2a1^EGFP* fetuses were used over four independent rounds of fluorescent-activated cell (FACS) sorting. Cell populations were isolated using the BD FACSAria Fusion with a 488 nm laser and a 530/30 filter for GFP. Initial FSC/SCC was set between 30/40 and 210/240 to exclude debris. After removal of dead cells with Draq7 and removal of doublets, following standard protocol, cells were gated for sorting.

### Flow cytometry

GFP signal presented in Supplementary Fig. 12B, 12D was quantified with a Beckman Coulter CytoFLEX flow cytometer using a 488 nm laser and a 525/40 filter. Single cells were identified based on their FSC/SCC features and data was formatted using *FlowJo* v10.9.

### ATAC-seq, ChIP-seq and RNA-seq processing

**ATAC-seq.** After sorting, cells were centrifuged for 5' at 400 × *g* at 4 °C and supernatant was discarded. Two biological replicates of 7.5 × 10⁴ cells for each condition were then isolated for direct processing with Nextera Tn5 enzyme (Illumina, FC-131-1096). Samples were treated as previously described[85] with the exception that elution of the Tn5 treated DNA was performed using the Monarch PCR and DNA clean-up kit (NEB T1030S) following a 5:1 ratio of Binding Buffer:Sample. We then build libraries using customized NextSeq primers and Kapa HiFi HotStart ReadyMix (Roche, 07958927001). Libraries were sequenced on an Illumina HiSeq 4000 with paired-end reads of 50 bases.

**ChIP-seq.** Following FACS sorting, cells were centrifuged for 5' at 400 × *g* at 4 °C and supernatant was discarded. Cells for ChIP-seq were resuspended in 10% FCS/PBS and fixed in 1% formaldehyde. The fixation was blocked by the addition of 1.25 M glycine, cells were isolated by centrifugation (1000 × *g*, at 4 °C for 8'), resuspended in cold lysis buffer (10 mM Tris, pH 7.5, 10 mM NaCl, 5 mM MgCl₂, 0.1 mM EGTA, Protease Inhibitor (Roche, 04693159001)) and incubated on ice for 10' to isolate the cell nuclei. The nuclei were isolated by centrifugation

(1000 × *g*, at 4 °C for 3'), washed in cold 1× PBS, and stored frozen at −80 °C. 5 × 10⁵ fixed nuclei were sonicated to a 200–500 bp length with the Bioruptor Pico sonicator (Diagenode). H3K27Ac ChIP (Diagenode, C15410174) was performed as previously described[86], using 1/500 dilution of the antibody, with the addition of 5 mM of Na-Butyrate to all buffers. Libraries were prepared with <10 ng quantities of ChIP-enriched DNA as starting material and processed with the Illumina TruSeq ChIP kit according to manufacturer specifications. Libraries were sequenced on an Illumina HiSeq 4000 with single-end reads of 50 or 100 bases.

**RNA-seq.** After sorting, cells were centrifuged for 5' at 400 × *g* at 4 °C, supernatant was discarded, and cells frozen at −80 °C. Two biological replicates of 1.5 × 10⁵ cells each were used to extract total RNA using RNeasy Micro Kit (Qiagen, 74004) for each condition. SMART-Seq v4 kit (Clontech, 634893) was used for the reverse transcription and cDNA amplification according to the manufacturer's specifications, starting with 5 ng of total RNA as input. 200 pg of cDNA were used for library preparation using Nextera XT (Illumina, FC-131-1096). Libraries were sequenced on an Illumina HiSeq 4000 with single-end reads of 50 bases. For the bulk samples, cells were directly frozen at −80 °C following dissociation. Four biological replicates of 2 × 10⁵ cells were used to extract total RNA using RNeasy Micro Kit (Qiagen, 74004) for each condition. Stranded mRNA Prep Ligation kit (Illumina, 20040534) was used starting from 100 ng of total RNA as input. Libraries were sequenced on an Illumina NovaSeq 6000 with single-end reads of 50 bases.

### ATAC-seq, ChIP-seq and RNA-seq analysis

**ATAC-seq.** ATAC-seq datasets were analyzed following[87]. NextSeq adapter sequences and bad quality bases were removed using *CutAdapt* v1.18[88] (-a CTGTCTCTTATACACATCTCCGAGCCCACGAGAC -A CTGTCTCTTATACACATCTGACGCTGCCGACGA -q30 -m15). Reads were then mapped to GRCm39/mm39 using *Bowtie2* v2.3.5.1[89] (--very-sensitive --no-unal --no-mixed --no-discordant --dovetail -X 1000). Reads with mapping quality below 30, mapping to mitochondria, or not properly paired were removed from the analysis with *Samtools view* v1.10[90]. PCR duplicates were filtered using *Picard* v2.21.1 (https://github.com/broadinstitute/picard). BAM file was converted to BED with *Bedtools* v2.28.0[91]. Peak calling and coverage was done using *MACS2* v2.2.7.1[92] (callpeak --nomodel --call-summits --extsize 200 --shift -100 --keep-dup all). Coverage was normalized by the number of millions of reads falling into *MACS2* summits +/− 500 bp using *Bedtools* v2.28.0[91]. When indicated, coverage profiles represent an average of the normalized coverage of all replicates.

**ChIP-seq.** A similar number of reads (39×10⁶) were randomly sampled from each ChIP-seq dataset to correct for sequencing depth variation using *Seqtk* v1.3 (-s 100) (https://github.com/lh3/seqtk). TruSeq adapter sequences and bad quality bases were removed using *CutAdapt* v1.18[88] (-a AGATCGGAAGAGCACACGTCTGAACTCCAGTCAC -q30 -m15). Reads were then mapped to GRCm39/mm39 using *Bowtie2* v2.3.5.1[89] with default parameters. Reads were then filtered for a MAPQ ≥ 30 using *Samtools view* v1.10[90] and the coverage and peak calling was obtained after extension of the reads by 200 bp using *MACS2* v2.2.7.1[92] (callpeak --nomodel --call-summits --extsize 200). Coverage was normalized by the number of million tags used by *MACS2*.

**RNA-seq.** For cDNA libraries generated from FACS sorted cells, NextSeq adapter sequences and bad quality bases were removed using *CutAdapt* v1.18[88] (-a CTGTCTCTTATACACATCTCCGAGCCCACGAGAC -q30 -m15). Unstranded reads were then mapped to both GRCm39/mm39 (for coverage visualization presented in Fig. 4, Supplementary Fig. 8 and 9) and to a customized GRCm39/mm39_eGFP-SV40pA for

differential gene expression analysis. We used *STAR* v2.7.2b[93] with the filtered GTFs (see Genomic data section) allowing to get a gene quantification simultaneously (--outSAMstrandField intronMotif --sjdbOverhang '99' --sjdbGTFfile $gtfFile --quantMode GeneCounts --outFilterType BySJout --outFilterMultimapNmax 20 --out-FilterMismatchNmax 999 --outFilterMismatchNoverReadLmax 0.04 --alignIntronMin 20 --alignIntronMax 1000000 --alignMatesGapMax 1000000 --alignSJoverhangMin 8 --alignSJDBoverhangMin 1). Gene expression computations were performed using uniquely mapping reads extracted from STAR alignments and genomic annotations from filtered GTF (see Genomic data section). FPKM values were determined by *Cufflinks* v2.2.1[94] (--max-bundle-length 10000000 --multi-read-correct --library-type "fr-unstranded" --no-effective-length-correction -M MTmouse.gtf). Coverage was computed with *Bedtools* v2.28.0[91] using uniquely mapped reads (NH:i:1 tag). When indicated, coverage profiles represent an average of the replicates. This was done by dividing each replicate by the number of millions of uniquely mapped reads (for normalization) and calculating the average coverage. Differentially expressed genes were tested using the R package *DESeq2* v1.34.0[95]. Chondrogenic (EGFP + ) and non-chondrogenic (EGFP-) marker genes were defined from the *DESeq2* Wald test results with thresholds being respectively set at ($\log_2$FC)>1.5 and ($\log_2$FC)<−1.5 with FDR-corrected by Benjamini-Hochberg method p-value < 0.05. Bulk stranded cDNA libraries were processed with the following differences. TruSeq adapter sequences instead of Nextera sequences were removed using *CutAdapt* (-a AGATCGGAAGAGCACACGTCTGAACTCCAGTCAC -q30 -m15). At the mapping step, the option --outSAMstrandField intronMotif was removed and the library type was set to "fr-firststrand" instead of "fr-unstrand" in *Cufflinks*. For differentially expressed genes no threshold was applied on the $\log_2$FC.

**Genomic tracks visualizations.** ATAC-seq, ChIP-seq and RNA-seq tracks presented in the figures were generate using *pyGenomeTracks* v3.8[96,97].

**Identification of enhancers.** *Bedtools merge* and *intersect* v2.30.0[91] and *deeptools multiBigwigSummary* v3.5.1[98] were used to identify putative enhancer. Briefly, for each tissue EGFP+ H3K27ac MACS2 narrowPeak peaks were intersected with EGFP + ATAC peaks present in both duplicates and extended by 150 bp on each side. ChIP peaks of interest were then filtered against a −2kb/+500b window centered at transcription start sites of protein coding genes to exclude promoters and proximal *cis* regulatory elements. Coverage on filtered ChIP peaks of interest was computed with *deepTools multiBigwigSummary* for the normalized coverage of H3K27ac ChIP of EGFP+ and EGFP-. Putative chondrogenic enhancers were then called as intervals displaying a fold change of normalized H3K27ac coverage in EGFP + /EGFP- ≥4 and an EGFP+ normalized H3K27ac ChIP coverage ≥0.5 and were then merged within 500 bp using *Bedtools merge*. A reciprocal analysis was followed to identify non-chondrogenic enhancers (EGFP-/EGFP+ ≥4 and an EGFP-normalized H3K27ac ChIP coverage ≥0.5). Chondrogenic and non-chondrogenic enhancers are listed respectively in Supplementary Data 5 and 6. To determine chondrogenic enhancer tissue specificity (Supplementary Data 5), the chondrogenic enhancers identified in each tissue were aggregated (overlapping peaks were merged). The normalized H3K27ac coverage was computed with *deepTools multiBigwigSummary* and a two-fold enrichment between coverages was used as a threshold to characterize these peaks into limb-enriched, trunk-enriched, and *pan*-chondrogenic enhancers. Chondrogenic enhancers were lifted from mm39 to hg38 using the UCSC *liftover* tool (https://genome.ucsc.edu/cgi-bin/hgLiftOver) (Supplementary Data 20).

**Categorization of ChondroTADs and ChondroEnhTADs.** Raw reads from Hi-C datasets generated from E14.5 mouse forelimb cartilage[29] were downloaded from SRA (SRP339920) and mapped to mm39 using *HiCUP* v0.8.1[99], *Samtools* v.1.10[90] and *Cooler*

v0.9.3[100]. The BAM file was then converted to a tabular file with a Python script (https://github.com/lldelisle/tools-lldelisle/blob/8ac44b0341c70ce330fc0f24712b6f9b59b14731/tools/fromHicupToJuicebox/fromHicupToJuicebox.py). The mapped read pairs were then loaded to 20-kb resolution matrices with *Cooler makebins*. The two raw matrices replicates were merged using *HiCExplorer* v3.7.2[97,101] *hicSumMatrices* and normalized with *Cooler balance* --cis-only. TADs were then called using *hicFindTADs* --minDepth 650000 --maxDepth 1300000 --step 1300000. Overlap between chondrogenic enhancers, protein-coding chondrogenic genes and TADs was performed using the R packages *GenomicRanges* v1.48.0[102] and *plyranges* v1.16.0[103]. *ChondroTADs* and *chondroEnhTADs* coordinates are listed in Supplementary Data 7 and 10.

**Gene Ontology enrichment analysis.** Protein-coding genes located in *chondroTAD* (Supplementary Data 12) and *chondroEnhTAD* (Supplementary Data 14) were analyzed for GO biological process enrichment using the PANTHER Overrepresentation Test (Released 20231017) using the website http://geneontology.org[104,105].

**Motif enrichment analysis.** We performed our motif enrichment analyses using the R package *motifcounter* v1.18.0[33]. This method relies on a higher-order Markov background model to compute the expected motif hits and a compound Poisson approximation for testing the motif enrichment compared to the chosen background. We use the default parameters for the order of the background model (order=1) and the false-positive level for motif hits ($\alpha = 0.001$). In our analysis, we apply the method to two sets of genomic regions[1], chondrogenic enhancers and[2] accessible regions without H3K27ac signal ("*inactive*"). In both sets, the genomic regions were centered at the corresponding ATAC-seq peaks. The same genomic region sets were used as background, respectively. After obtaining the enrichment scores for both sets, we refer to the $\log_2$ fold-change value for the over- or under-representation of a motif in the chondrogenic enhancers compared to the "*inactive*" regions. All genomic regions were reduced to a length of 500 bp before the analysis. We tested for enrichment of the binding profiles of 356 TFs in total which were downloaded from the HOCOMOCO database (mouse core collection, v11, mononucleotide PCMs)[106].

**Heatmap visualizations.** Read signals for ChIP-seq and ATAC-seq data were visualized with the *plotHeatmap* function of *deepTools* v3.5.1[107]. The represented genomic regions were centered on the corresponding ATAC-seq peaks, where proximal peaks (center less than 75 bp apart) were merged to avoid redundancy, if indicated in figure caption.

**Comparisons with published experiments**
**Mouse SOX9 ChIP-seq.** E14.5 SOX9 ChIP-seq fastq reads were obtained from ref. 40 and reprocessed using the analysis pipeline described above. MACS2 narrowPeak peaks were merged within 500 bp (Supplementary Data 16) and then overlapped with our chondrogenic enhancer set using *Bedtools merge* and *intersect* v2.30.0[91].

**Mouse ATAC-seq.** ATAC-seq fastq reads were obtained for E9.75, E10.5 and E11.5 from[32] and for E11.5 to E14.5 from[31] and reprocessed using the analysis pipeline described above. The latter datasets were produced by the Axel Visel and Len Pennacchio laboratory, LBNL with following identifiers ENCSR377YDY (E11.5), ENCSR551WBK (E12.5), ENCSR896XIN (E13.5), ENCSR460BUL (E14.5).

**Mouse longshanks.** We lifted the TAD spans of the eight discrete murine *Longshanks* loci reported by Castro et al[42]. from mm10 to mm39 using the UCSC *liftover* tool (https://genome.ucsc.edu/cgi-bin/

hgLiftOver). Overlap between these eight regions and our chondrogenic enhancers and chondrogenic protein-coding genes was then performed using the R packages *GenomicRanges* v1.48.0[102] and *plyranges* v1.16.0[103]. The circos plot was produced using the R package *circlize* v0.4.15[108].

**Human GWAS height.** We analyzed the overlap of our enhancer sets with published human GWS-loci[44], which were defined as non-overlapping genomic segments that contain at least one quasi-independent genome-wide significant (GWS) SNP associated to adult human height as well as common SNPs from the HapMap3 project in the close vicinity of GWS SNPs. We lifted 7'209 GWS-loci from the EUR cohort from hg19 to mm10 using the UCSC *liftover* tool (https://genome.ucsc.edu/cgi-bin/hgLiftOver), resulting in 6'926 GWS loci (283 conversions failed). Next, we lifted these regions over to mm39, resulting in the final 6'916 GWS loci we used for our overlap analyses (10 conversions failed). For some analyses, we split the GWS loci into "protein-coding" and a "non-protein-coding" GWS loci based on their overlap with 21'614 protein-coding genes obtained from a filtered GTF file based on ENSEMBL GRCm39 release 104 (see Genomic data section for details). We found 5'145 GWS loci overlapping with at least one protein-coding gene, and 1'771 GWS loci without overlap. For each GWS-loci, we also knew the genetic variance of height explained by GWS SNPs within the loci. The overlap analysis of enhancers and GWS-loci was done using the *findOverlaps* function of the R package *GenomicRanges* v1.38.0[102]. Specifically, we checked for each GWS-loci in an iterative manner, starting from the one with the highest variance explained, if there are any overlapping enhancers. Then, we assign the corresponding variance to the overlapping enhancer(s) and plot the cumulative variances of GWS-loci (x-axis) against the corresponding cumulative variance that can be explained by overlapping enhancers (y-axis). The enhancer sets were matched in size by sorting enhancers based on the maximum H3K27ac enrichment in limb or trunk, and considering only the top 877 enhancers.

**Enhancer assay**
Transgenic E14.5 mouse fetuses were generated as described previously[109]. Briefly, super-ovulating female FVB mice were mated with FVB males and fertilized embryos were collected from the oviducts. Regulatory element sequences (Supplementary Data 21) were amplified from human genomic DNA or synthesized (Twist Biosciences) and cloned into the donor plasmid containing a minimal β-globin promoter, *lacZ* reporter gene and H11 locus homology arms using NEBuilder HiFi DNA Assembly Mix (NEB, E2621). The sequence identity of donor plasmids was verified using Nanopore sequencing (Primordium Labs). Plasmids are available upon request. A mixture of Cas9 protein (Alt-R SpCas9 Nuclease V3, IDT, Cat#1081058, final concentration 20 ng/μL), hybridized sgRNA against H11 locus (Alt-R CRISPR-Cas9 tracrRNA, IDT, cat#1072532 and Alt-R CRISPR-Cas9 locus targeting crRNA, gctgatggaacaggtaacaa, total final concentration 50 ng/μL) and donor plasmid (12.5 ng/μL) was injected into the pronucleus of donor FVB embryos. The efficiency of targeting and the gRNA selection process is described in detail in ref. 110. Embryos were cultured in M16 with amino acids at 37°C, 5% CO2 for 2 h and implanted into pseudopregnant CD-1 mice. Fetuses were collected at E14.5 for *lacZ* staining as described previously[109]. Briefly, fetuses were dissected from the uterine horns, washed in cold PBS, fixed in 4% PFA for 30 min and washed three times in embryo wash buffer (2 mM MgCl2, 0.02% NP-40, and 0.01% deoxycholate in PBS at pH 7.3). They were subsequently stained overnight at room temperature in X-gal stain (4 mM potassium ferricyanide, 4 mM potassium ferrocyanide, 1 mg/mL X-gal and 20 mM Tris pH 7.5 in embryo wash buffer). PCR using genomic DNA extracted from embryonic sacs digested with DirectPCR Lysis Reagent (Viagen, 301-C) containing Proteinase K (final concentration 6 U/mL) was used to confirm integration at the H11 locus and test for presence of tandem insertions[109]. Only fetuses with donor plasmid insertion at H11 were used. Fetuses of both sexes were used in the analysis. The stained transgenic fetuses were washed three times in PBS and imaged from both sides using a Leica MZ16 microscope and Leica DFC420 digital camera. All images are in the Vista enhancer browser (https://enhancer.lbl.gov/)[110] with the hs numbers 2696, 2697, 2698 and 2700.

**Reporting summary**
Further information on research design is available in the Nature Portfolio Reporting Summary linked to this article.

## Data availability
Sequencing data are available in the GEO repository under the accession number GSE230235. Mouse E13.5 limb SOX9 ChIP-seq was obtained from the SRA website [https://www.ncbi.nlm.nih.gov/sra/] with accession number DRX028798. Mouse embryonic limb ATAC-seq datasets were obtained from the GEO repository [https://www.ncbi.nlm.nih.gov/geo/query/acc.cgi?acc=GSE164738] and the SRA website [https://www.ncbi.nlm.nih.gov/sra/] with the following accession numbers: SRR14305872, SRR14305873, SRR14305249, SRR14305250, SRR14305866, SRR14305867, SRR14306149, SRR14306150, SRR14306037 and SRR14306038. The customized GRCm39/mm39_eGFP-SV40pA sequence and the corresponding filtered GTF file are available on Zenodo [https://doi.org/10.5281/zenodo.7837435]. Source data are provided with this paper.

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

## Acknowledgements
We thank Mylène Docquier, Brice Petit, Didier Chollet and Christelle Barraclough from the iGE3 sequencing facility. We thank Grégory Schneiter, Lan Tran and Cécile Gameiro from the Flow Cytometry facility. We thank Olivier Fazio, Angélique Vincent and Fabrizio Thorel from the Transgenic facility. We thank Nicolas Liaudet from the Bioimaging facility and Stéphane Pàges, Laura Batti and Ivana Gantar from Advanced Light Sheet Imaging Center (ALICe) at the Wyss Center for Bio and Neuroengineering, Geneva. We thank Jennifer Akiyama from LBNL for careful imaging of the LacZ transgenic fetuses. We thank Jozsef Zakany, Timothy Frayling and all Andrey lab members for discussions and critical reading of the manuscript. The computations were performed at University of Geneva using Baobab HPC service. This study was supported by grants from the Swiss National Science Foundation PP00P3_176802, PP00P3_210996, from the Von Meissner Foundation and from a donator advised by CARIGEST to G.A as well as the Swiss National Science Foundation grant P400PB_194334 to F.D. L.L.-D. was supported by the European Research Council (ERC) #588029 and by EPFL. Research conducted at the E.O. Lawrence Berkeley National Laboratory was performed under U.S. Department of Energy Contract DE-AC02-05CH11231, University of California. A.V. and M.K. were supported by NIH grants R01DE028599 and R01HG003988.

## Author contributions
G.A. conceived the project. F.D., Ant.R., and Z.J. performed the mESC targetings, fetus imaging, ATAC-seq, ChIP-seq, scRNA-seq and RNA-seq preparations. F.D., Ann.R., and L.L.-D. performed computational analyses. M.K. performed transgenic analyses. G.A., F.D., and A.V. wrote the manuscript with input from the remaining authors.

## Competing interests
The authors declare no competing interests.
