## [Peer Review File · Nature Communications]

Pre-Hypertrophic Chondrogenic Enhancer Landscape of Limb and Axial Skeleton DevelopmentReviewers' Comments:

Reviewer #1:

Remarks to the Author:

The authors generated the reporter mice, which express EGFP under the control of enhancers of Col2a1 gene locus. They collected EGFP+ and EGFP- cells from limbs and trunk at E14.5, ATAC seq and H3K27ac modification were examined in EGFP+ and EGFP- cells, and chondrogenic and non-chondrogenic enhancers were defined. Further, they performed motif analysis of transcription factors in the enhancer candidates, and had consistent results with a previously reported Sox9 ChIP in the H3K27ac+ active enhancers. They compared the loci related to tibial length in mice with those of enhancer candidates, and the chondrogenic enhancers were enriched in the loci related to tibial length. Finally, they found that human height loci were overlapped with chondrogenic enhancer regions. Further, the chondrogenic enhancers were matched with the previously reported enhancers of the genes involved in chondrogenesis. They also confirmed chondrocyte-specific expression of the reporter mice using chondrogenic enhancers of Col2a1 and Fgfr3 by themselves. The experiments were well designed and performed straightforward. Although they systematically examined and the data may be useful to find the enhancers of height-related genes, most results shown here are already known and lack novel findings and insights.

Specific comment:

Which parts of limbs and trunk were used for the isolation of EGFP+ and EGFP- cells? It should be described in detail.

Reviewer #2:

Remarks to the Author:

In the manuscript titled "Chondrogenic Enhancer Landscape of Limb and Axial Skeleton Development," the authors conducted RNA-Seq, ATAC-Seq, and H3K27Ac ChIP-Seq profiling on limb and trunk fetal chondrocytes isolated from E14.5 mouse embryos to identify chondrogenic genes and enhancers. The authors demonstrated that the identified chondrogenic enhancers are associated with chondrogenic genes, and their activities can be attributed to the binding of key transcription factors, such as Sox9. Additionally, the authors established a link between chondrogenic enhancers and human height. Overall, the data presented in this study appear to be of high quality. The enhancer atlases generated in the study are a valuable addition to the fields of skeleton development and enhancer biology. However, it should be noted that several previously published studies have attempted to characterize the cis-regulatory landscapes in mouse chondrocytes. It would be beneficial for the authors to comment on and compare their findings with those from previous studies.

My major concern is that the study is overall descriptive and lacks functional validation. It is crucial to validate the functions of chondrogenic enhancers, particularly those associated with human height loci, in regulating chondrogenic gene expression and chondrogenesis using cell-based assays and ideally animal models.

Specific points:

1. The current title of the manuscript may be slightly misleading and could benefit from modification. In this study, the authors focused exclusively on characterizing enhancers in E14.5 Col2a1-positive cells, which represent early to pre-hypertrophic chondrocytes, rather than mature hypertrophic chondrocytes. Therefore, the current study did not characterize the complete enhancer landscape during the developmental course of limb and axial skeleton development.
2. In this study, the authors constructed a "Col2a1 fluorescent regulatory sensor" by inserting an EGFP reporter upstream of the TSS of the Col2a1 gene. Theoretically, this new reporter system should provide a more accurate representation of the endogenous Col2a1 expression pattern compared to the

transgene-based Col2a1-ECFP reporter line (Chokalingam et al., 2009, PMID: 19231914), which has been widely used for isolating chondrocytes. It would be nice if the authors could directly compare the reporter expression pattern of the two lines and demonstrate the superior performance of the Col2a1 fluorescent regulatory sensor.

3. ATAC-Seq has been performed on Col2a1-ECFP-positive rib chondrocytes and Col10a1-mCherry-positive rib hypertrophic chondrocytes in newborn mice (Hojo et al., Cell Rep 2022, PMID: 36070691), and long bone chondrocytes from E15.5 mouse embryos (Richard et al., Cell 2020, PMID: 32220312). The authors should discuss why they think the chondrogenic enhancer atlases generated in the current study are more accurate and/or functionally relevant.

4. Line 190: 18% and 8% of chondrogenic enhancers are limb-specific or trunk-specific. What may be the causes of these tissue-specific enhancer activities? Do limb-specific or trunk-specific enhancers show enrichment of certain transcription factor binding motifs? Or could these tissue-specific activities be attributed to the slightly different developmental timing between trunk and limb cartilages?

5. Line 196: The authors used TAD annotation from mouse ESC to analyze the regulatory relationships between chondrogenic enhancers and chondrogenic genes. Given that the TAD organization may vary between different cell and tissue types, the authors should use Hi-C data of developing limbs for this analysis.

6. Line 200-203: The authors categorized TADs as ChondroTAD and ChondroEnhTAD to evaluate whether a chondrogenic enhancer regulates a chondrogenic gene or a general developmental gene (i.e. non-chondrogenic gene). The authors could consider utilizing strategies like the ABC model (Fulco et al., 2019, PMID: 31784727) to deduce enhancer-gene connections, provided that suitable Hi-C datasets are available. This approach could enhance the downstream analyses and further strengthen the findings of the study."

7. Line 266: Sox9 only binds to 39% of chondrogenic enhancers. Are Sox9-bound enhancers enriched for those regulating chondrogenic genes (i.e. chondrogenic enhancers located in chondroTADs)?

8. Related to Figure 4B. Are enhancers more likely to be causal variants than non-enhancers? The non-coding variants not overlapping with any enhancers should be included as a control.

9. Related to Figure 4C, D. As the author stated, the main value of the chondrogenic enhancer atlas is that it provides a framework for identifying causal genetic variations. The authors have shown by analyses that the genetic variation overlapping chondrogenic enhancers can explain human height variations. It is natural to expect functional validation of these important enhancers. The author should delete and/or modify some of the chondrogenic enhancers associated with the human height loci using CRISPR/ CRISPRi, and examine the functional consequences of the enhancer perturbations on target gene expression and chondrogenesis.

Minor points

Figure S4A: EGFP gene is labeled as "EGFP" in Limb (top left) and "Egfp_SV40pA" in Trunk (top right). Also, the positions for the panels are indicated wrongly in the figure legend. (trunk (bottom left) and combined datasets (top right)).

REVIEWER COMMENTS

Reviewer #1 (Remarks to the Author):

The authors generated the reporter mice, which express EGFP under the control of enhancers of Col2a1 gene locus. They collected EGFP+ and EGFP- cells from limbs and trunk at E14.5, ATAC seq and H3K27ac modification were examined in EGFP+ and EGFP- cells, and chondrogenic and non-chondrogenic enhancers were defined. Further, they performed motif analysis of transcription factors in the enhancer candidates, and had consistent results with a previously reported Sox9 ChIP in the H3K27ac+ active enhancers. They compared the loci related to tibial length in mice with those of enhancer candidates, and the chondrogenic enhancers were enriched in the loci related to tibial length. Finally, they found that human height loci were overlapped with chondrogenic enhancer regions. Further, the chondrogenic enhancers were matched with the previously reported enhancers of the genes involved in chondrogenesis. They also confirmed chondrocyte-specific expression of the reporter mice using chondrogenic enhancers of Col2a1 and Fgfr3 by themselves.

The experiments were well designed and performed straightforward. Although they systematically examined and the data may be useful to find the enhancers of height-related genes, most results shown here are already known and lack novel findings and insights.

We are glad the reviewer found the experiments well-designed and sorry that she/he did not find our insights novel. We hope that with the new data and revised manuscript, in particular with the novel in vivo functional assessment of enhancers presented in the new Fig. 5, this reviewer will change his opinion.

Specific comment:

Which parts of limbs and trunk were used for the isolation of EGFP+ and EGFP- cells? It should be described in detail.

In Fig. 1A legend line 162, we have now written: "right: sketch of microdissected limb and trunk tissues. In particular, stylo-, zeugo-, and autopods were harvested from fore and hindlimbs. Trunks were isolated from neck to tail, and internal organs were removed prior to processing" We have also added a direct reference to this scheme in the text at line 134.

Reviewer #2 (Remarks to the Author):

In the manuscript titled "Chondrogenic Enhancer Landscape of Limb and Axial Skeleton Development," the authors conducted RNA-Seq, ATAC-Seq, and H3K27Ac ChIP-Seq profiling on limb and trunk fetal chondrocytes isolated from E14.5 mouse embryos to identify chondrogenic genes and enhancers. The authors demonstrated that the identified chondrogenic enhancers are associated with chondrogenic genes, and their activities can be attributed to the binding of key transcription factors, such as Sox9. Additionally, the authors established a link between chondrogenic enhancers and human height. Overall, the data presented in this study appear to be of high quality. The enhancer atlases generated in the study are a valuable addition to the fields of skeleton development and enhancer biology. However, it should be noted that several previously published studies have attempted to characterize the cis-regulatory landscapes in mouse chondrocytes. It would be beneficial for the authors to comment on and compare their findings with those from previous studies.

My major concern is that the study is overall descriptive and lacks functional validation. It is crucial to validate the functions of chondrogenic enhancers, particularly those associated with human height loci, in regulating chondrogenic gene expression and chondrogenesis using cell-based assays and ideally animal models.

We generally thank the reviewer for his/her valuable comments. To address his/her major concern about the functional validation of our enhancer set, we engineered and characterized enhancer deletions in vivo. In particular, we measured the effect of four homozygous deletions of chondrogenic enhancers overlapping major height-associated loci (Hhip1, Nkx3-2, Fgfr3, and Col2a1) and assay in limb and trunk the induced transcriptional effects.

Specific points:

1. The current title of the manuscript may be slightly misleading and could benefit from modification. In this study, the authors focused exclusively on characterizing enhancers in E14.5 Col2a1-positive cells, which represent early to pre-hypertrophic chondrocytes, rather than mature hypertrophic chondrocytes. Therefore, the current study did not characterize the complete enhancer landscape during the developmental course of limb and axial skeleton development.

Following this reviewer's suggestion, we have now changed the title of the manuscript to: "Pre-Hypertrophic Chondrogenic Enhancer Landscape of Limb and Axial Skeleton Development". We are also now precisising in the Result and in the Discussion sections that we map pre-hypertrophic chondrogenic enhancers (lines 155 and 449).

2. In this study, the authors constructed a “Col2a1 fluorescent regulatory sensor” by inserting an EGFP reporter upstream of the TSS of the Col2a1 gene. Theoretically, this new reporter system should provide a more accurate representation of the endogenous Col2a1 expression pattern compared to the transgene-based Col2a1-ECFP reporter line (Chokalingam et al., 2009, PMID: 19231914), which has been widely used for isolating chondrocytes. It would be nice if the authors could directly compare the reporter expression pattern of the two lines and demonstrate the superior performance of the Col2a1 fluorescent regulatory sensor.

To answer the reviewer’s important point, we exploited Hojo et al. Cell Rep 2022 (PMID: 36070691) to compare RNA-seq data between Col2a1-ECFP P1 rib chondrocytes and our EGFP-sorted Col2a1^{EGFP} sensor E14.5 trunk chondrocytes. Differential gene expression highlighted consistency between both datasets, yet differences emerged in their bias towards pre-hypertrophic versus hypertrophic chondrocytes. Our dataset emphasized pre-hypertrophic markers like *Matn1*, *Matn4*, *Ihh*, *Nkx3-2*, and *Igfbp5*, while the Hojo dataset favored more differentiated, hypertrophic markers such as *Spp1* and *Col10a1* (see Figure 1 below for mentioned genes and their ratios to Col2a1 in the table below). Yet, distinguishing the impact of sampled stages (E14.5 vs P1) versus the transgene effect remains challenging. Given the absence of a comparable Col2a1-ECFP transcriptomic dataset, creating one for this manuscript lies beyond its scope.

Figure 1: A. Genes displaying a significant expression preference in ECFP-sorted cells isolated from *tg(Col2a1-eCFP)* P1 ribs (purple, Hojo et al., Cell 2022, PMID: 36070691) versus EGFP-sorted cells isolated from *Col2a1^{EGFP}* sensor E14.5 trunk (orange). Statistical test used: DESeq2 Wald test, significant enrichment was scored when $abs(\log_2FC) > 1.5$ and FDR-corrected $p\text{-value} < 0.05$. **B.** Ratios of *Spp1* and *Col10a1* over *Col2a1* in the two datasets considered.

On top, the *Col2a1^{EGFP}* sensor approach has two major advantages listed below:

- The *Col2a1-ECFP* transgene (Tsumaki et al., 1999, PMID: 9885252, Chokalingam et al., 2009, PMID: 19191514 and 19231914) is deprived of several chondrogenic enhancers, notably *hs2697* and *hs2698* described in this work. When these enhancers are absent (see response to point 9), *Col2a1* expression in the trunk is reduced, suggesting that both enhancers provide an important regulatory input. It therefore cannot be excluded that the transcription the *Col2a1-ECFP* transgene does not bear all regulatory features.
- The *Col2a1^{EGFP}* sensor uses an mESC-based set-up directly compatible with our own tetraploid aggregation pipeline, already used in Rouco et al., 2021, PMID: 34903763. This approach has the advantage of greatly reducing the number of animals used for such experiments.

As integrating all analyses in the revised version is beyond its scope, we refer to our sensor versus the *Col2a1-ECFP* transgene specificity by writing at line 443: “In contrast to a previously described *Col2a1* transgene-based approach (59-61), this EGFP knock-in bears all the regulatory specificities of the *Col2a1* locus. This is particularly

visible as the loss of two Col2a1 distal enhancers, that are not included in the transgene-based approach, showed a significant reduction of Col2a1 and EGFP expression in the trunk. Generally, this result showcases the importance of a complete regulatory landscape for driving accurate reporter gene expression.”.

3. ATAC-Seq has been performed on Col2a1-ECFP-positive rib chondrocytes and Col10a1-mCherry-positive rib hypertrophic chondrocytes in newborn mice (Hojo et al., Cell Rep 2022, PMID: 36070691), and long bone chondrocytes from E15.5 mouse embryos (Richard et al., Cell 2020, PMID: 32220312). The authors should discuss why they think the chondrogenic enhancer atlases generated in the current study are more accurate and/or functionally relevant.

With our approach, the H3K27ac coverage, on top of ATAC-seq, allows us to distinguish active enhancers from poised ones (Rada-Iglesias et al., 2011, PMID: 21160473). We therefore identify a much smaller but stringent set of enhancers with a decreased false positive rate. To further illustrate this point, and as suggested by this reviewer, we performed a series of comparisons between our datasets and the Hojo et al. and the Richard et al. publications. We first compared with our dataset the open chromatin regions found in Hojo et al. Col2a1-eCFP sorted P1 ribs ATAC-seq data and E15.5 distal and proximal femur by Richard et al. Here, we reprocessed the data in our own analysis pipeline for better comparison. We measured 93'592 Hojo and 18'463 Richard open regions located outside of TSS outside of promoter (against 112'095 in our own dataset) and compared them with our 2'704 chondrogenic enhancers. We found that 80% of our enhancers (2'250/2'704) overlap the Hojo dataset and 64% (1'745/2'704) the Richard dataset. In both cases, our enhancer category was a few percent of the total open regions. This is very similar to our own analysis, where we found that 72% of open regions do not have H3K27ac and that 25% bear the mark but not specifically in chondrocytes. We therefore believe that H3K27ac is a true added value to map tissue-specific enhancer elements among the vast amount of open chromatin regions.

In the Hojo dataset, differences in ATAC-seq signals obtained from different sub-chondrogenic cell types are used to further identify cell-type specific chondrogenic enhancers. We therefore compared this list of regions with our set of enhancers. Specifically, we found, after lifting over in mm39 and removing regions overlapping TSS, that 874 regions are found in cells expressing only Col2a1, while 2'017 regions are found in cells expressing Col2a1 but also Col10a1 or Sp7. Only 50 of our elements overlapped the first category while 211 the second. This limited overlap shows that basing all profiling on chromatin accessibility identifies a different enhancer set than when adding other enhancer-associating marks. Yet, we also notice a difference in the development stage between Hojo (P1) and our E14.5 dataset.

As for the previous comment, we believe integrating all these analyses in the revised version is beyond its scope. Yet, we summarize its main and general aspects writing:

Line 84: “Finally, the open chromatin signatures of chondrocytes in vivo were mapped to characterize their epigenetic landscape, yet, with limited specificity when it comes to identifying active chondrogenic enhancer regions as such a signature alone both marks poised and active regions (11, 12).”

Line 451: “The usage of H3K27 acetylation to define active enhancers in our dataset explains the reduced number of high-confidence enhancers compared to other studies based solely on chromatin accessibility (11, 12, 28).”

4. Line 190: 18% and 8% of chondrogenic enhancers are limb-specific or trunk-specific. What may be the causes of these tissue-specific enhancer activities? Do limb-specific or trunk-specific enhancers show enrichment of certain transcription factor binding motifs? Or could these tissue-specific activities be attributed to the slightly different developmental timing between trunk and limb cartilages?

To answer the referee point, we have measured whether limb- and trunk-enriched enhancers displayed differential transcription factor motifs in a new Supplementary Figure S8B. We now refer to the result line 268: “We then measured whether limb- and trunk-enriched enhancers displayed differential transcription factor motifs and only observed a limited enrichment of PRXX2 motif in the limb-enriched enhancers compared to the trunk-enriched ones (Supplementary Fig. S8B).”

5. Line 196: The authors used TAD annotation from mouse ESC to analyze the regulatory relationships between chondrogenic enhancers and chondrogenic genes. Given that the TAD organization may vary between different cell and tissue types, the authors should use Hi-C data of developing limbs for this analysis.

We thank this reviewer for this suggestion. We have now incorporated in our analysis a HiC dataset generated from a biologically relevant cell type (E14.5 limb cartilage, Chen et al. 2022, PMID: 36417512). We have recalculated all the enhancer attributions following this new dataset. The only substantial difference that came along with this change is that chondrogenic enhancer from chondroEnhTADs overlaps now with a bit more cumulative height variance than ones from chondroTADs (see Fig. 4B). In the previous version, both types of enhancers overlapped with the same cumulative height variance.

6. Line 200-203: The authors categorized TADs as ChondroTAD and ChondroEnhTAD to evaluate whether a chondrogenic enhancer regulates a chondrogenic gene or a general developmental gene (i.e. non-chondrogenic gene). The authors could consider utilizing strategies like the ABC model (Fulco et al., 2019, PMID: 31784727) to deduce enhancer-gene connections, provided that suitable Hi-C datasets are available. This approach could enhance the downstream analyses and further strengthen the findings of the study."

*We thank this reviewer for this suggestion and have run the ABC model on our (limb) genomic datasets in combination with the Chen et al. E14.5 limb cartilage HiC. However, a major pitfall of this tool is that it is designed to link not enhancers to gene(s) but genes to enhancers. We believe this distinction to be extremely important as ABC computed an average number of 37 TSS allocated to every single chondrogenic enhancer investigated (limb-enriched and pan-chondrogenic enhancers, n=2'487). Furthermore, those connections are stretching over vast genomic distances (average of 2.03 ± 1.43 Mb), not always compatible with the accepted functional partition of the genome in TADs. While we agree that single enhancers might, in particular cases, control more than one single target gene, such situations are rare and observed at very particular genomic positions, such as gene clusters. Moreover, our impression was further reinforced by the observation that upon chondrogenic enhancers deletion (see point 9), differentially expressed genes in cis (i.e., *Nkx3-2*, *Hhip*, *Col2a1*, and *Fgfr3*) were preferentially located within the enhancer TAD. We therefore conclude that our current TAD-based analysis, while biased and limited to a certain extent, is nevertheless biologically relevant.*

7. Line 266: Sox9 only binds to 39% of chondrogenic enhancers. Are Sox9-bound enhancers enriched for those regulating chondrogenic genes (i.e. chondrogenic enhancers located in chondroTADs)?

We thank this reviewer for this suggestion. We computed the numbers and now write line 278: "We did not observe a differential proportion of SOX9-bound enhancers in chondroTADs (38.7%, 498/1'287) and chondroEnhTADs (39%, 458/1'175)."

8. Related to Figure 4B. Are enhancers more likely to be causal variants than non-enhancers? The non-coding variants not overlapping with any enhancers should be included as a control.

Although we understand the "spirit" of the referee's first part of the question, it is difficult to answer it in a straightforward way as we cannot define "non-enhancer" regions as any region not defined in this study could bear a regulatory potential in another context/tissue. These other regions could actually be enhancers regulating other physiological processes linked to height (for instance, growth hormone) or even enhancers active in other steps of bone formation. However, to pinpoint the non-overlapping height variance part and to answer the second part of the comment, we included the fraction of coding and non-coding variants overlapping chondrogenic enhancers (and therefore the fraction not overlapping chondrogenic enhancers) in a new Supplementary Table S9 and at:

Line 319: "To do so, we computed the variance explained by the overlap between the 6'916 mouse-conserved height variance-explaining loci (mHVEL) accounting for 42.5% of height variance with chondrogenic enhancers. We observed that chondrogenic enhancers overlapped 1'293 (18.6%) of them and could provide an interpretation framework for 24.4% of mHVEL variance (Supplementary Table 11). Yet, as most height variance-explaining loci contain both coding and non-coding segments, the variance explained by the overlap of mHVEL with chondrogenic enhancer is also accounting for variations within coding parts of genes, and particularly chondrogenic genes. Therefore, to focus on variation occurring at non-coding segments only we decided to further focus our analysis on the 1'771 mHVEL deprived of any protein-coding gene (non-coding mHVEL). Cumulatively, non-coding mHVEL explain 5.7% of the height variance while the 293 (14.6%) of them overlapping with chondrogenic enhancers account for 18.4% of this variance (Supplementary Table 11). We then aimed at measuring whether chondrogenic enhancer are more likely to explain height variance than other enhancer regions."

Line 512: “Here, we showed that between 18% (considering only non-coding mHVEL) to 24% (considering all mHVEL) of the height variance explained overlapped chondrogenic enhancers, while the rest of the variance could be associated either to regions involved in other parts of bone formation and to regions linked to height variance but unrelated to chondrogenesis.”

9. Related to Figure 4C, D. As the author stated, the main value of the chondrogenic enhancer atlas is that it provides a framework for identifying causal genetic variations. The authors have shown by analyses that the genetic variation overlapping chondrogenic enhancers can explain human height variations. It is natural to expect functional validation of these important enhancers. The author should delete and/or modify some of the chondrogenic enhancers associated with the human height loci using CRISPR/ CRISPRi, and examine the functional consequences of the enhancer perturbations on target gene expression and chondrogenesis.

To answer this important point, we engineered the deletion of four predicted chondrogenic enhancer regions to discern their role in controlling their cognate gene in developing limbs and trunks. Notably, deletions at the Col2a1 and Fgfr3 loci demonstrated substantial impacts, with the removal of chondrogenic enhancers resulting in significant reductions in the expression of Col2a1 and Fgfr3 genes. Further examination extended to enhancer regions at the Hhip and Nkx3-2 loci, revealing specific limb reductions in Hhip transcripts and expression losses in both limbs and trunk for Nkx3-2. These functional insights showcase the utility of the genome-wide set of predicted chondrogenic enhancers in unraveling the regulatory landscape of individual loci and establishing connections between human genetic variation, gene regulation, and stature. This part of the revision yielded a new Figure 5 and Supplementary Figure S12 and associated text section starting at line 385.

Minor points

Figure S4A: EGFP gene is labeled as “EGFP” in Limb (top left) and “Egfp_SV40pA” in Trunk (top right). Also, the positions for the panels are indicated wrongly in the figure legend. (trunk (bottom left) and combined datasets (top right)).

We thank the reviewer for spotting these errors. We have now corrected them.

Reviewers' Comments:

Reviewer #1:

Remarks to the Author:

Authors added the Fig. 5 and showed the physiological significance of the chondrogenic enhancers in limb and trunk, which they systematically selected by mRNA seq, ATAC seq, and ChIP seq (H3K27ac) using the EGFP reporter mice in Col2a1 locus. It sufficiently improved the paper.

Reviewer #2:

Remarks to the Author:

I commend the efforts by the authors to address the issues I raised during the initial review, particularly by conducting functional validation of chondrogenic enhancers using multiple enhancer deletion mouse models. While the concept that tissue-specific enhancers can act at a distance to regulate gene expression is not entirely new, this study does provide important resources for understanding the transcriptional regulatory mechanisms during skeletal development and the genetic underpinnings for human stature variations and bone diseases. I think the revised manuscript is overall suitable for publication in Nature Communications.

Several minor issues that require attention:

1. In line 207, the authors mention that the 661 chondroEnhTADs harbor 1315 chondrogenic enhancers, corresponding to ~ 2 enhancers per TAD, rather than 1.3 enhancers per TAD as shown on line 209. This needs to be corrected and the statistical analysis needs to be redone.
2. Figure 2D: each dot in this box plot corresponds to the number of enhancers in a TAD, yet on the figure the names of developmental genes contained by the TAD were indicated. This needs to be more clearly explained in the figure legend.
3. In lines 340-341, the observation that chondroEnhTAD enhancers can account for a greater variance in phenotypes compared to chondroTAD enhancers is intriguing. It suggests that the regulatory influence of general developmental genes by chondrogenic enhancers may be equally, if not more, functionally significant than the regulation of chondrogenic genes by chondrogenic enhancers. This point should be articulated more clearly either within the results section or during the discussion.
4. Line 395: it should be hs2698 instead of hs2898.

REVIEWER COMMENTS

Reviewer #1 (Remarks to the Author):

Authors added the Fig. 5 and showed the physiological significance of the chondrogenic enhancers in limb and trunk, which they systematically selected by mRNA seq, ATAC seq, and ChIP seq (H3K27ac) using the EGFP reporter mice in Col2a1 locus. It sufficiently improved the paper.

We thank this reviewer for his/her positive perception of the revised version of our manuscript.

Reviewer #2 (Remarks to the Author):

I commend the efforts by the authors to address the issues I raised during the initial review, particularly by conducting functional validation of chondrogenic enhancers using multiple enhancer deletion mouse models. While the concept that tissue-specific enhancers can act at a distance to regulate gene expression is not entirely new, this study does provide important resources for understanding the transcriptional regulatory mechanisms during skeletal development and the genetic underpinnings for human stature variations and bone diseases. I think the revised manuscript is overall suitable for publication in Nature Communications.

We thank this reviewer for his/her positive perception of the revised version of our manuscript and provide answers below to the minor issues raised by this reviewer.

Several minor issues that require attention:

1. In line 207, the authors mention that the 661 chondroEnhTADs harbor 1315 chondrogenic enhancers, corresponding to ~2 enhancers per TAD, rather than 1.3 enhancers per TAD as shown on line 209. This needs to be corrected and the statistical analysis needs to be redone.

We thank this reviewer for spotting this error that propagated from the first version of the manuscript. We have now corrected at line 195 the average number of chondrogenic enhancers located in chondroEnhTADs (1.99 rounded up to 2) in the manuscript and have repeated the Wilcoxon rank sum test. We can confirm that the result of the statistical test remains unchanged with $W = 76018$ and $p\text{-value} < 2.2e-16$.

2. Figure 2D: each dot in this box plot corresponds to the number of enhancers in a TAD, yet on the figure the names of developmental genes contained by the TAD were indicated. This needs to be more clearly explained in the figure legend.

We have now included an additional sentence in the Figure 2D legend, line 1036, to clarify this point: "Dots represent the number of chondrogenic enhancers contained in each TAD, and some relevant examples are named based on the genes they encompass."

3. In lines 340-341, the observation that chondroEnhTAD enhancers can account for a greater variance in phenotypes compared to chondroTAD enhancers is intriguing. It suggests that the regulatory influence of general developmental genes by chondrogenic enhancers may be equally, if not more, functionally significant than the regulation of chondrogenic genes by chondrogenic enhancers. This point should be articulated more clearly either within the results section or during the discussion.

We have now stated directly in the result section line 303: "This suggests that height variants affecting chondrogenic enhancers, which control general developmental genes, may be equally or even more functionally significant than those controlling chondrogenic genes."

4. Line 395: it should be hs2698 instead of hs2898.

The typo was corrected (line 340).

Reviewers' Comments:

Reviewer #2:

Remarks to the Author:

The authors fully addressed my concerns and the manuscript is ready for publication.